# Causal Attention to Exploit Transient Emergence of Causal Effect

## Abstract

We propose a causal reasoning mechanism called *causal attention* that can improve performance of machine learning models on a class of causal inference tasks by revealing the generation process behind the observed data. We consider the problem of reconstructing causal networks (e.g., biological neural networks) connecting large numbers of variables (e.g., nerve cells), of which evolution is governed by nonlinear dynamics consisting of weak coupling-drive (i.e., causal effect) and strong self-drive (dominants the evolution). The core difficulty is sparseness of causal effect that emerges (the coupling force is significant) only momentarily and otherwise remains quiescent in the neural activity sequence. *Causal attention* is designed to guide the model to make inference focusing on the critical regions of time series data where causality may manifest. Specifically, attention coefficients are assigned autonomously by a neural network trained to maximise the Attention-extended Transfer Entropy, which is a novel generalization of the iconic transfer entropy metric. Our results show that, without any prior knowledge of dynamics, *causal attention* explicitly identifies areas where the strength of coupling-drive is distinctly greater than zero. This innovation substantially improves reconstruction performance for both synthetic and real causal networks using data generated by neuronal models widely used in neuroscience.

## 1 Introduction

In this work, our task is to infer causal relationships between observed variables based on time series data and reconstruct the causal network connecting large numbers of these variables. Assume the time series $x_{it}$ record the time evolution of variable $i$ governed by coupled nonlinear dynamics, as represented by a general differential equation $x_{it} = g(x_{it}) + \sum B_{ij} f(x_{it}, x_{jt})$, where $g$ and $f$ are self- and coupling functions respectively. The parent variable influences the dynamic evolution of the child variable via the coupling function $f$. Note that these two functions are hidden and usually unknown for real systems. The asymmetric adjacency matrix $B$ represents the causal, i.e., directional coupling relationship between variables. Hence, the goal is to infer matrix $B$ from observed time series $x_{it}, i = 1, 2, \ldots, N$ where $N$ is the number of variables in the system. If $B_{ij} = 1$, the variable $i$ is a coupling driver (parent variable) of variable $j$, otherwise it is zero.

The key challenge is that the causal effect in neural dynamics (e.g., biological neural systems observed via neuronal activity sequences) is too weak to be detected, rendering powerless classic unsupervised techniques of causal inference across multiple research communities Granger (1969); Schreiber (2000); Sugihara et al. (2012); Sun et al. (2015); Nauta et al. (2019); Runge et al. (2019); Gerhardus & Runge (2020); Tank et al. (2021); Mastakouri et al. (2021). This difficulty manifests in three aspects. First, the dynamics contains self-drive and coupling-drive. The strength of coupling $f(x_{it}, \cdot)$ is usually many orders of magnitude smaller than self-drive $g(x_{it})$, and the latter dominates evolution. Second, the behavior of the coupling-drive is chaotic, unlike in linear models Shimizu et al. (2006); Xie et al. (2020). The resulting unpredictability and variability of system state means that coupling force can be significant momentarily and otherwise almost vanish, as illustrated in Figure 3 (gray lines). This dilutes the information in time series that can be useful for inferring the causal relationship. Third, in the heterogeneous networks common in applications, some variables are hubs coupled with many parent variables, among which it is difficult to distinguish individual causes. When causal effects are weak, we do not observe clearly the principle of Granger Causality, whereby the parent variable can help to explain the future change in its child variable Pfister et al.

(2019). Rather, when we train a machine learning model Nauta et al. (2019); Tank et al. (2021) for prediction task on the neuronal activity sequences, the model only exploits the historical information of the child variable itself and that from parent variables is ignored. We posit that coupling-drive makes a negligible contribution to dynamic evolution in the majority of samples of time series data. In other words, only in a small fraction of samples is the information of parent variables effective in predicting the evolution of child variables. Taking as an example the gradient algorithm to minimise the regression error over all samples $\sum_t (x_{it} - \hat{x}_{it})^2$, the adjustment of model parameters from the tiny samples corresponding to significant coupling force is negligible, but these are the only samples which could induce the model to exploit causal effects in reducing regression error. Similarly, for transfer entropy Schreiber (2000), which measures the reduction in uncertainty which a potential parent variable provides to a potential child variable, there is no significant difference in measured value between ordered pairs of variables with and without causality.

To overcome the difficulty, we propose a causal reasoning mechanism – *causal attention* – to identify the moments when causal effect emerges. We design an objective function, **A**ttention-**e**xtended **T**ransfer **E**ntropy (**AeTE**), comprising a weighted generalisation of transfer entropy. In order to maximize **AeTE**, the causal attention mechanism trains neural networks to autonomously allocates high attention coefficients $a_t$ at times $t$ where information of parent variables effectively reduces the uncertainty of child variables, and ignores other positions by setting $a_t$ close to zero. If we consider each value in a time series as a feature, the operation of attention allocation is also equivalent to removing the non-causal features Kusner et al. (2017); Hu et al. (2021).

However, noise in empirical samples may also produce high transfer entropy regions, which leads to spurious causal effects even when using causal attention. We add a binary classification model to perform more sophisticated inference under the guidance of causal attention to focus on these critical regions and recognize different patterns between noisy and sparse emergence of causal effect. We deal with this class of causal inference task by way of small sample supervised learning. Although training and test data have a distribution shift in the setting of small samples, they arise through an identical underlying generation process. Thus, if the model provides an insight into the underlying dynamics – the coupling-drive for causal inference – then the understanding acquired from small samples can be effectively utilised in the test environment Bareinboim & Pearl (2014); Battaglia et al. (2016); Makhlouf et al. (2020); Pessach & Shmueli (2022). The role of causal attention is to help the classification model gain this insight. Our contributions are summarized as follows:

1. We introduce *causal attention*, a causal reasoning mechanism to identify the positions of time series at which causal effect emerges and guide a classification model to infer causality focusing on these critical positions. Without any prior knowledge of dynamics, the mechanism determines the areas where the coupling force is substantially different from zero.

2. By formulating Transfer Entropy as the difference between two types of mutual information, and based on the dual representation of Kullback-Liebler (KL) divergence, we design a differentiable metric, Attention-extended Transfer Entropy, as the objective function of the proposed causal attention mechanism.

3. Our method significantly improves performance on synthetic and real causal networks using the data generated by five well-known neural dynamic models, and the number of labels required is very small compared to the size of the causal networks.

Our methodology has limitations (i.e., cases for which performance improvement is less): 1. Dense networks, where a variable is coupled with many driving variables such that their causal effects overlap and are harder to distinguish. 2. Intense noise, which makes the casual attention mechanism falsely identify high transfer entropy regions. The downstream classifier then extracts non-causal features, leading to the reduction of its generalization. 3. Strongly coupled system, which is dominated by synchronization phenomena in which the dynamic behaviors of all variables are similar.

## 2 BACKGROUND

### 2.1 DEFINITION OF TRANSFER ENTROPY

The transfer entropy, an information-theoretic causality measure, is able to detect information flow between time series $X$ and $Y$. Transfer Entropy measures the degree of non-symmetric dependence

of $Y$ on $X$, defined as Schreiber (2000):

$$TE(X \to Y) = \sum p\left(y_{t+1}, y_t^{(k)}, x_t^{(l)}\right) \log \frac{p\left(y_{t+1} \mid y_t^{(k)}, x_t^{(l)}\right)}{p\left(y_{t+1} \mid y_t^{(k)}\right)}, \tag{1}$$

where $x_t^{(l)} = (x_t, ..., x_{t-l+1})$ and $y_t^{(k)} = (y_t, ..., y_{t-k+1})$ and $k$, $l$ are lengths. For an uncoupled system ($X$ and $Y$ independent) that can be approximated by a Markov process of order $k$, the conditional probability to find $Y$ in state $y_{t+1}$ at time $t + 1$ satisfies $p\left(y_{t+1} \mid y_t^{(k)}, x_t^{(l)}\right) = p\left(y_{t+1} \mid y_t^{(k)}\right)$.

## 2.2 MUTUAL INFORMATION NEURAL ESTIMATION

The mutual information is equivalent to the KL divergence between the joint distribution $P_{XY}$ and the product of the marginal distributions $P_X \otimes P_Y$ Nowozin et al. (2016); Hjelm et al. (2018). The KL divergence $D_{KL}$ admits the neural dual representation Donsker & Varadhan (1983); Belghazi et al. (2018):

$$MI(X, Y) = D_{KL}\left(P_{XY} \| P_X, P_Y\right) \geq \sup_{\theta \in \Theta} E_{P_{XY}}\left[f_\theta\right] - \log\left(E_{P_X \otimes P_Y}\left[e^{f_\theta}\right]\right), \tag{2}$$

where the supremum is taken over parameter space $\Theta$ and $f_\theta$ is the family of functions parameterized by the neural network with parameters $\theta \in \Theta$. The mutual information neural estimator is strongly consistent and can approximate the actual value with arbitrary accuracy Belghazi et al. (2018).

## 3 METHOD

### 3.1 NEURAL ESTIMATOR OF TRANSFER ENTROPY

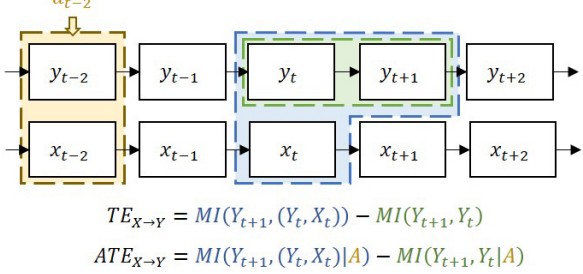

$$TE_{X \to Y} = MI(Y_{t+1}, (Y_t, X_t)) - MI(Y_{t+1}, Y_t)$$
$$ATE_{X \to Y} = MI(Y_{t+1}, (Y_t, X_t)|A) - MI(Y_{t+1}, Y_t|A)$$

Figure 1: Visual interpretation of transfer entropy and its attention-extended version. The Transfer Entropy is derived as the difference of two types of mutual information: $MI\left(Y_{t+1}, (Y_t, X_t)\right)$ (blue area) quantifies the reduction in uncertainty of future state $y_{t+1}$ from knowing current states $(y_t, x_t)$, and $MI\left(Y_{t+1}, Y_t\right)$ (green area) is same but only $y_t$ is known. The attention coefficients $a_t$ (yellow area) are assigned to each position of time series by the *causal attention* mechanism to maximize the Attention-extended Transfer Entropy. For brevity, $k = l = 1$ here.

By the conditional Bayes formula and adding a marginal distribution of $Y$, we derive the transfer entropy as the difference between two types of mutual information. An intuitive description is provided in Figure 1, and the derivation is placed in Appendix A.

$$TE(X \to Y)$$

$$= \sum p\left(y_{t+1}, y_t^{(k)}, x_t^{(l)}\right) \log \frac{p\left(y_{t+1}, y_t^{(k)}, x_t^{(l)}\right)}{p\left(y_r\right) p\left(y_t^{(k)}, x_t^{(l)}\right)} - \sum p\left(y_{t+1}, y_t^{(k)}\right) \log \frac{p\left(y_{t+1}, y_t^{(k)}\right)}{p\left(y_r\right) p\left(y_t^{(k)}\right)} \tag{3}$$

$$= MI\left(Y_{t+1}, \left(Y_t^{(k)}, X_t^{(l)}\right)\right) - MI\left(Y_{t+1}, Y_t^{(k)}\right). \tag{4}$$

In these expressions, $y_r$ is sampled from $\mathbf{Y}$ randomly and independently of the time step $t$. The first term $MI\left(Y_{t+1}, \left(Y_t^{(k)}, X_t^{(l)}\right)\right)$ quantifies the reduction in the uncertainty of the future state $y_{t+1}$ from knowing the historical information $y_t^{(k)}$ and $x_t^{(l)}$. The second term $MI\left(Y_{t+1}, Y_t^{(k)}\right)$ is the reduction in uncertainty simply from knowing $y_t^{(k)}$. By connecting Eq. 4 and Eq. 2, we define a differentiable estimator of transfer entropy as:

$$TE(X \to Y) = \sup_{\Theta} E_{P\left(Y_{t+1}, Y_t^{(k)}, X_t^{(l)}\right)}[f_\theta] - \log\left(E_{P(Y_{t+1}) \otimes P\left(Y_t^{(k)}, X_t^{(l)}\right)}\left[e^{f_\theta}\right]\right)$$
$$- \sup_{\Phi} E_{P\left(Y_{t+1}, Y_t^{(k)}\right)}[f_\phi] - \log\left(E_{P(Y_{t+1}) \otimes P(Y_t^{(k)})}\left[e^{f_\phi}\right]\right). \tag{5}$$

Transfer entropy, and even mutual information, is difficult to compute Paninski (2003), especially for high-dimensional or noisy data. In Appendix B, we offer a theoretical proof for the consistency and convergence properties of Transfer Entropy Neural Estimation, and examine its bias on a linear dynamic system where the true values of transfer entropy can be determined analytically.

### 3.2 ATTENTION-EXTENDED TRANSFER ENTROPY

The main difficulty in our task is that the causal effect in certain nonlinear dynamical systems is too weak to be recognized by classic techniques. We discuss the limitation of the iconic transfer entropy in detail that it works well when the three true distributions, i.e., one joint distribution and two conditional distributions in Eq. 1, can be estimated perfectly. However, sparse causal effects are easily masked if the estimated probability density deviates even slightly from the real distribution. These momentary sources of evidence of coupling drive are like outliers in the total distribution of a time series dominated by self-drive. In order to make the transfer entropy provide a clear distinction between causal and non-causal pairs, we need to highlight the positions where $p\left(y_{t+1} \mid y_t^{(k)}, x_t^{(l)}\right) > p\left(y_{t+1} \mid y_t^{(k)}\right)$ and filter out other times by adjusting $a_t$ in Eq. 6, all while avoiding the problem of distribution approximation. We do so by defining **AeTE** as:

$$AeTE(X \to Y) = \sum a_t \cdot p\left(y_{t+1}, y_t^{(k)}, x_t^{(l)}\right) \log \frac{p\left(y_{t+1} \mid y_t^{(k)}, x_t^{(l)}\right)}{p\left(y_{t+1} \mid y_t^{(k)}\right)} \tag{6}$$

$$= MI\left(Y_{t+1}, \left(Y_t^{(k)}, X_t^{(k)}\right) \mid A\right) - MI\left(Y_{t+1}, Y_t^{(k)} \mid A\right). \tag{7}$$

In this expression, $a_t \in [0, 1]$ is the attention coefficient at time step $t$ and the collection $A$ of attention coefficients is the attention series. Comparison of Eq.1 and Eq.6 reveals that the transfer entropy can be viewed as a simplified version of **AeTE** in which attention coefficients are uniformly set to one: $\forall t, a_t = 1$. Because each position has an equal contribution to estimation, the value of transfer entropy is dominated by the majority of positions where causal effect is negligible, i.e., where $p\left(y_{t+1} \mid y_t^{(k)}, x_t^{(l)}\right) \approx p\left(y_{t+1} \mid y_t^{(k)}\right)$. Similarly to transfer entropy, **AeTE** is derived as the difference of two mutual informations, but **AeTE** incorporates the scheme of attention assignment. By connecting Eq. 7 and Eq. 2, we define a differentiable estimator of **AeTE** as:

$$AeTE(X \to Y) = \sup_{\Theta} \frac{1}{L} \sum a_t \cdot f_\theta\left(y_{t+1}, y_t^{(k)}, x_t^{(l)}\right) - \log\left(\frac{1}{L} \sum a_t \cdot e^{f_\theta\left(y_r, y_t^{(k)}, x_t^{(l)}\right)}\right)$$
$$- \sup_{\Phi} \frac{1}{L} \sum a_t \cdot f_\phi\left(y_{t+1}, y_t^{(k)}\right) - \log\left(\frac{1}{L} \sum a_t \cdot e^{f_\phi\left(y_r, y_t^{(k)}\right)}\right), \tag{8}$$

where $T$ is the total number of time steps and $L = T - \max(k, l)$. The expectation on the distribution of variables is adapted into the mean over time series.

### 3.3 CAUSAL ATTENTION MECHANISM

The overall framework of our model is presented in Figure 2. In addition to two neural networks $f_\theta$ and $f_\phi$ for mutual information estimation, we employ another neural network $g_\alpha$ for causal attention

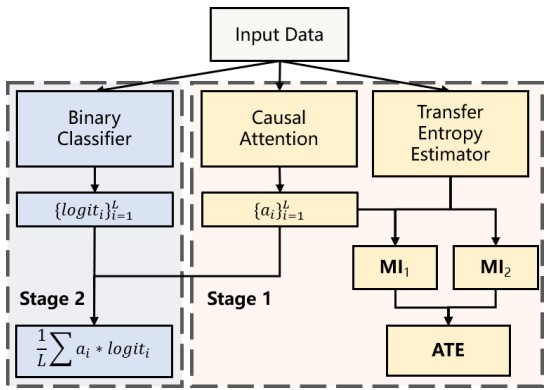

Figure 2: Graphical illustration of causal attention mechanism framework. An input sample is the time series on an order pair of variables with shape $[2, L]$. Stage 1: the neural network $g_\alpha$ assigns the attention coefficients $\{a_i\}_{i=1}^L$. The neural networks $f_\theta$ and $f_\phi$ forming transfer entropy estimator estimate mutual information $MI_1$ and $MI_2$ (first and second terms in Eq. 7). Stage 2: the inferred probability of causality is $Sigmoid(\frac{1}{L}\sum a_i * logit_i)$.

assignment. Rather than approximating distributions, the neural network $g_\alpha$ learns to maximize **AeTE** given by Eq. 8 via gradient descent. However, the occurrence of high transfer entropy regions may appear due to noise in empirical samples. For more sophisticated inference, we augment our method with a binary classifier $h_\eta$ guided by causal attention to focus on high transfer entropy regions and recognize different patterns between noise and sparse emergence of causal effect. The classifier takes causality and non-causality as class labels. Then, the training process is divided into two independent stages: causal attention learning and classification learning. The objectives in the first stage are:

$$\theta, \phi \leftarrow \underset{\theta, \phi | \alpha}{\mathrm{argmax}} \; \mathcal{L}_1 + \mathcal{L}_2 \tag{9}$$

$$\alpha \leftarrow \underset{\alpha | \theta, \phi}{\mathrm{argmax}} \; \mathcal{L}_1 - \mathcal{L}_2, \tag{10}$$

where $\mathcal{L}_1, \mathcal{L}_2$ is the expectation of the first and second sup term of Eq.8 on training set respectively. We update $(f_\theta, f_\phi)$ and $g_\alpha$ alternately. A small learning rate is required to maintain training stability, otherwise the $g_\alpha$ may fall into a trivial solution where attention is almost zero throughout the time series. The objective in the second stage is:

$$\eta \leftarrow \underset{\eta | \alpha}{\mathrm{argmin}} \; \mathcal{L}_3, \tag{11}$$

where $\mathcal{L}_3$ is the binary cross entropy and the notation $\eta \mid \alpha$ indicates that causal attention remains fixed during the second stage of training. The downstream classifier is sensitive to the upstream scheme of attention assignment. In experiments, there exists an optimal loss interval for training the attention model $g_\alpha$. We stop the first stage training when the loss value of objective Eq. 10 is stable in this interval, and then the downstream classifier $h_\eta$ usually obtains the best generalization performance. For different dynamics, their optimal intervals are different and we find them empirically[1]. Details on the implementation of causal attention mechanism are provided in Appendix C.

## 4 EXPERIMENT

We describe our experiment setup and extensively evaluate the causal attention mechanism on neuronal dynamics coupled on both synthetic and real causal networks.

### 4.1 SETUP

**Causal networks.** For synthetic networks, we generate ten groups of Erdős-Rényi (ER) and scale-free (SF) directed networks with one hundred nodes (i.e., variables) uniformly and with mean degree varying from 5 to 41 by adjusting the probability for edge creation in ER and the total number of

---

[1]An alternative design, which we have not yet implemented, would involve joining the first and second stages. Attention model $g_\alpha$ would be trained by not only maximizing AeTE but also responding to feedback from the classifier, and would find the balance between Eq. 10 and Eq. 11 automatically.

edges in SF. Symmetric links (both $x_i \rightarrow x_j$ and $x_j \rightarrow x_i$) can exist. For each set of network parameters we consider five independently generated instances. For real networks, we select five neurological connectivity datasets as presented in Table 1, each from a different species: Cat, Macaque, Mouse, Worm and Rat.

| Dataset | Region | #Nodes | #Edges | Mean degree |
|---------|--------|--------|--------|-------------|
| Cat | Brain | 65 | 1139 | 17.5 |
| Macaque | Brain | 242 | 4090 | 16.9 |
| Mouse | Cortex | 195 | 214 | 1.1 |
| Worm | Neural | 272 | 4451 | 16.4 |
| Rat | Brain | 503 | 30088 | 59.8 |

Table 1: Statistical information of five real networks: dataset name, type of network, number of nodes, number of edges and mean degree $\langle k \rangle$. Detailed introduction are provided in Appendix E.

Table 2: Equations of the five dynamical models considered. $B$ is the asymmetrical adjacency matrix of the causal network, recording causal relationships between nodes. $B_{ij} = 1$ if variable $i$ is the parent of variable $j$, otherwise $B_{ij} = 0$. In these expressions, $\Gamma$ describes the coupling-drive, while other terms represent self-drive. The detailed configuration of dynamical parameters is provided in Appendix D.

| Dynamics | Equations |
|----------|-----------|
| Hindmarsh-Rose | $\dot{p}_i = q_i - ap_i^3 + bp_i^2 - n + I_{\text{ext}} + \Gamma$ 
 $\dot{q}_i = c - dp_i^2 - q_i, \quad \dot{n}_i = r\left[s\left(p_i - p_0\right) - n_i\right]$ 
 $\Gamma = g_c\left(V_{\text{syn}} - p_i\right)\sum_{j=1}^{N} B_{ij}/(1 + \exp(-\lambda\left(p_j - \Theta_{\text{syn}}\right)))$ |
| Morris-Lecar | $C\dot{V} = I - g_L\left(V - V_L\right) - g_{\text{Ca}}m_\infty(V)\left(V - V_{\text{Ca}}\right) - g_K n\left(V - V_K\right) + \Gamma$ 
 $\dot{n} = \lambda(V)\left(n_\infty(V) - n\right), \quad \Gamma = g_c\sum_{j=1}^{N} B_{ij}\left(n_j - n_i\right)$ |
| Izhikevich | $\dot{v}_i = 0.04v_i^2 + 5v_i + 140 - u_i + I + \Gamma$ 
 $\dot{u}_i = a(bv_i - u_i), \quad \Gamma = g_c\sum_{j=1}^{N} B_{ij}u_j$ |
| Rulkov | $\text{F}_1\left(u_i, w_i\right) = \dfrac{\beta}{1 + u_i^2} + w_i + \Gamma\left(u_j\right), \quad \text{F}_2\left(u_i, w_i\right) = w_i - \nu u_i - \sigma$ 
 $\Gamma\left(u_j\right) = g_c\sum_{j=1}^{N} B_{ij}/\left(1 + \exp(\lambda\left(u_j - \Theta_s\right))\right)$ |
| FitzHugh-Nagumo | $\dot{v} = a + bv + cv^2 + dv^3 - u + \Gamma$ 
 $\dot{u} = \varepsilon(ev - u), \quad \Gamma = -g_c\sum_{j=1}^{N} B_{ij}\left(v_j - v_i\right).$ |

**Dynamic models.** We use five dynamic models for neural activity simulation widely used in the field of neuroscience: Hindmarsh-Rose (HR), Morris-Lecar (Morris), Izhikevich (Izh), Rulkov and FitzHugh-Nagumo (FHN). Dynamic equations are provided in Table 2, and segments of generated time series are represented in Figure 3.

**Evaluation metrics.** We measure the following metrics: (1) the area under the receiver operating characteristic curve (AUROC); and (2) the area under the precision-recall curve (AUPRC).

**Baselines.** We compared our method with seven baselines: (1) Granger causality test (Ganger) Granger (1969); (2) Transfer Entropy (TE), as in Eq. 5; (3) Convergent cross mapping (CCM) Sugihara et al. (2012); (4) Latent convergent cross mapping (Latent CCM) De Brouwer et al. (2020); (5) PCMCI Runge et al. (2019) and (6) PCMCI$^+$ Runge (2020) using partial correlation to quantify causal strength; (7) Classification model with Convolutional Block Attention Module (TA) Woo et al. (2018).

**Training details.** We employ the convolutional neural network for model $g_\alpha$ and $h_\eta$, and the fully-connected neural network for model $f_\theta$ and $f_\phi$. We use the ADAM Kingma & Ba (2014) optimizer with the initial learning rate of $10^{-4}$ for classifier $h_\eta$ and $10^{-5}$ for the others. The batch size is 10. For synthetic networks, we select randomly twenty ordered pairs of variables as a training/validation set and four hundred ordered pairs as a test set. For real networks, the sample set scheme is provided in Table 3. All sets are composed of equal samples with causality and without causality. The total time step $T$ of time series is 50,000. Gaussian measurement noise is added with mean zero and standard deviation $1\%/10\%$ that of the original time series for synthetic/real networks respectively. We run all experiments in this work on a local machine with two NVIDIA Tesla V100 32GB GPUs.

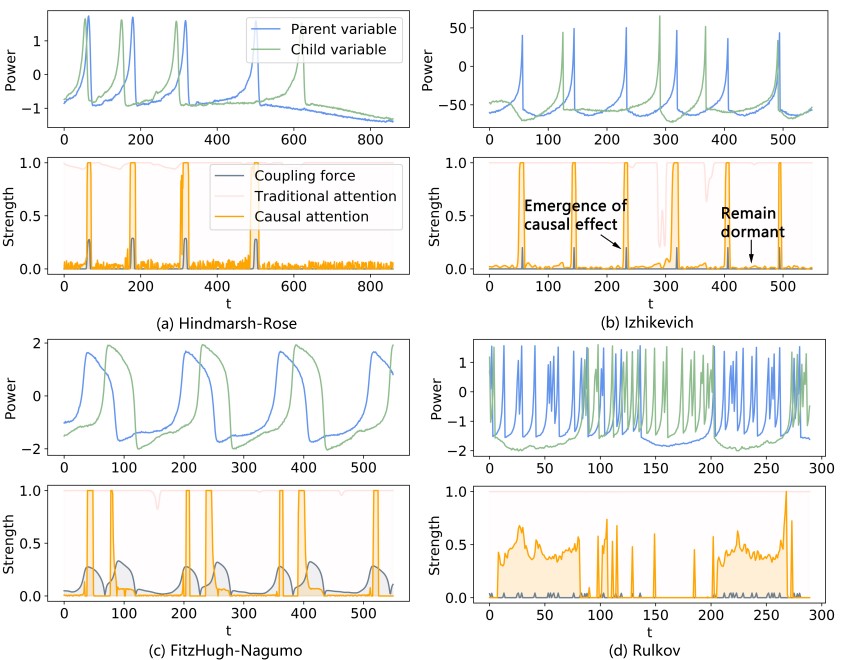

Figure 3: Insight into the coupling drive of dynamics. Top panel in each subplot: segment of time series from a ordered pair of variables with a causal relationship (blue is parent and green is child). Bottom panel in each subplot: the absolute value of coupling force (gray line), traditional attention (light pink line), and causal attention (orange line). (a) Hindmarsh-Rose; (b) Izhikevich; (c) FitzHugh-Nagumo; (d) Rulkov.

## 4.2 RESULT

**Insight into the coupling-drive of underlying dynamics.** The gray lines in Figure 3 represent the change of coupling force from parent to child variable over time, and are generated by the coupling term $\Gamma$ in Table 2. The absolute value of the coupling force rises (the gray lines spike) at occasional moments when the behavior of a parent variable substantially influences the evolution of its child variable, and remains almost zero at other times. The orange lines representing the causal attention keep in step with the gray lines, indicating that the causal attention mechanism recognizes the effect of coupling force in reducing the uncertainty of the child variable and pays attention to the areas where coupling force is significant. In Figure 3(d), the causal attention focuses on two separated regions where the coupling forces have concentrated bursts. In contrast, the light pink lines representing the traditional attention remain close to their maximum value, indicating it is insensitive to changes in coupling force. This leads its classifier to extract features throughout the whole time series (instead of focusing on causal features). The traditional attention are not designed for causal reasoning and cannot accommodate the selection of features that correspond to the causal information.

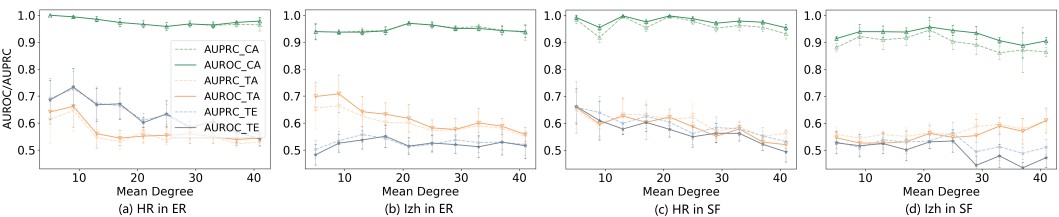

Figure 4: Comparison of classifiers on synthetic causal networks. CA: Causal Attention. Examples of dynamical model: (a) HR on ER; (b) Izh on ER; (c) HR on SF; (d) Izh on SF.

Table 3: Performance comparison on real causal networks. Values in the first column of each dataset are AUROC, and values in the second column are AUPRC. Each point contains the mean and standard deviation computed in five experiments with randomly sampled training/validation/test set in **Cat** (top, 10/10/500) and **Mouse** (bottom, 10/10/90) connectomes. Results on other three connectomes are shown in Appendix F.

| | Hindmarsh-Rose | | Morris-Lecar | | Izhikevich | | Rulkov | | FitzHugh-Nagumo | |
|---|---|---|---|---|---|---|---|---|---|---|
| **Granger** | 0.50±0.01 | 0.50±0.01 | 0.57±0.01 | 0.57±0.01 | 0.59±0.02 | 0.59±0.02 | 0.65±0.01 | 0.65±0.01 | 0.53±0.01 | 0.53±0.01 |
| **TE** | 0.59±0.01 | 0.54±0.01 | 0.44±0.01 | 0.49±0.01 | 0.40±0.03 | 0.44±0.02 | 0.74±0.01 | 0.73±0.02 | 0.57±0.02 | 0.62±0.01 |
| **CCM** | 0.75±0.01 | 0.75±0.02 | 0.58±0.02 | 0.58±0.02 | 0.51±0.01 | 0.51±0.01 | 0.57±0.01 | 0.57±0.01 | 0.68±0.02 | **0.68**±0.02 |
| **Latent CCM** | 0.69±0.02 | 0.69±0.01 | 0.55±0.02 | 0.55±0.02 | 0.48±0.03 | 0.48±0.03 | 0.51±0.01 | 0.51±0.02 | 0.64±0.02 | 0.64±0.01 |
| **PCMCI** | 0.53±0.01 | 0.53±0.01 | 0.49±0.01 | 0.49±0.01 | 0.49±0.01 | 0.49±0.01 | 0.66±0.01 | 0.66±0.01 | 0.53±0.01 | 0.53±0.01 |
| **PCMCI$^+$** | 0.57±0.01 | 0.57±0.01 | 0.52±0.02 | 0.52±0.01 | 0.52±0.02 | 0.52±0.01 | 0.63±0.02 | 0.63±0.02 | 0.56±0.01 | 0.56±0.01 |
| **TA** | 0.76±0.01 | 0.76±0.02 | 0.61±0.01 | 0.59±0.01 | 0.53±0.01 | 0.52±0.01 | 0.53±0.01 | 0.52±0.01 | 0.67±0.01 | 0.64±0.02 |
| **ATE** | **0.91**±0.01 | **0.88**±0.01 | **0.80**±0.01 | **0.76**±0.01 | **0.61**±0.02 | **0.62**±0.01 | **0.92**±0.01 | **0.89**±0.01 | **0.70**±0.01 | 0.67±0.01 |
| | Hindmarsh-Rose | | Morris-Lecar | | Izhikevich | | Rulkov | | FitzHugh-Nagumo | |
| **Granger** | 0.62±0.03 | 0.62±0.02 | 0.51±0.02 | 0.51±0.01 | 0.54±0.03 | 0.54±0.02 | 0.89±0.01 | 0.89±0.02 | 0.21±0.04 | 0.21±0.01 |
| **TE** | 0.40±0.03 | 0.44±0.02 | 0.53±0.05 | 0.62±0.08 | 0.57±0.02 | 0.56±0.03 | 0.33±0.04 | 0.40±0.02 | 0.30±0.03 | 0.33±0.01 |
| **CCM** | 0.55±0.05 | 0.55±0.04 | 0.24±0.05 | 0.24±0.01 | 0.47±0.04 | 0.47±0.04 | 0.58±0.01 | 0.58±0.02 | 0.58±0.05 | 0.58±0.04 |
| **Latent CCM** | 0.47±0.01 | 0.47±0.01 | 0.51±0.02 | 0.51±0.01 | 0.51±0.01 | 0.51±0.02 | 0.53±0.02 | 0.53±0.02 | 0.53±0.01 | 0.53±0.01 |
| **PCMCI** | 0.50±0.02 | 0.50±0.01 | 0.46±0.03 | 0.46±0.01 | 0.50±0.02 | 0.50±0.01 | 0.77±0.03 | 0.77±0.03 | 0.13±0.02 | 0.13±0.01 |
| **PCMCI$^+$** | 0.53±0.03 | 0.50±0.01 | 0.38±0.04 | 0.38±0.01 | 0.50±0.05 | 0.50±0.02 | 0.79±0.02 | 0.79±0.01 | 0.21±0.09 | 0.21±0.01 |
| **TA** | 0.97±0.02 | 0.95±0.04 | 0.51±0.02 | 0.53±0.02 | 0.51±0.03 | 0.51±0.02 | 0.89±0.01 | 0.83±0.05 | **0.94**±0.01 | **0.89**±0.01 |
| **ATE** | **0.98**±0.01 | **0.96**±0.02 | **0.85**±0.01 | **0.82**±0.01 | **0.92**±0.03 | **0.91**±0.03 | **0.98**±0.01 | **0.96**±0.02 | 0.89±0.04 | 0.85±0.05 |

**Performance on test sets.** Compared with the baselines, our method usually substantially improves reconstruction performance on both synthetic and real causal networks, as shown in Figure 4 and Table 3. In contrast, the classifier with traditional attention mechanism (TA) obtains low losses on training sets but has poor generalization on test sets, highlighting that mere statistical correlation for causal inference is unstable and can be spurious Cui & Athey (2022). The performance of classical unsupervised methods, for which all positions in the time series are treated equally, is also limited by the paucity of causal effects. These patterns demonstrate the importance of identifying and focusing on critical regions, which we achieve via the causal attention mechanism. In conclusion, our method slightly increases cost, due to the need for label collection, but obtains a substantial boost in performance compared with those unsupervised methods in this class of causal network reconstruction tasks.

The performance of all methods tends to decrease as the average network degree grows. Networks with larger average degree are more likely to exhibit synchronization of variables which makes it harder to distinguish cause and effect. Furthermore, a single variable in these denser networks can have many parent variables, and substantial coupling forces can emerge from distinct parents at overlapping times, making individual drivers harder to distinguish. In this circumstance, a slight variance in the scheme of causal attention assignment may cause fluctuations in the performance of the downstream classifier. Robustness of the proposed method to measurement noise and sequence length is presented in Appendix F.

## 5 RELATED WORK

### 5.1 CAUSAL NETWORK RECONSTRUCTION

Conventional frameworks assume separability, i.e., that information about causes are not contained in the caused variable itself. Several common methods Spirtes & Glymour (1991); Sun et al. (2015); Runge et al. (2019); Mastakouri et al. (2021) are based on conditional independence relations, but

differ in the design of condition-selection strategies or choice of conditional independence test. Granger Causality Granger (1969) is extended to nonlinear dynamics by using neural networks to represent nonlinear casual relationships Tank et al. (2021); Nauta et al. (2019). Many methods of causal discovery assume that the causal network is a directed acyclic graph. However, directed cyclic graphs are common in real systems. To address this non-separability issue, Convergent-cross mapping Sugihara et al. (2012) and its variations Clark et al. (2015); De Brouwer et al. (2020) measure the extent to which the historical record of child can reliably estimate states of the parent in reconstructed state space. However, sparse causal effect in neuronal dynamics, particularly in the presence of noise, may lead parent and child time series to appear statistically independent, so that their contribution to state estimation is hard to recognize.

## 5.2 MUTUAL INFORMATION ESTIMATION

Belghazi et al. Belghazi et al. (2018) built on a dual representation of KL divergence Donsker & Varadhan (1983) to offer a parametric mutual information neural estimator (MINE) which is linearly scalable in dimension as well as sample size, and is also trainable and strongly consistent. They also discussed another version of MINE based on the $f$-divergence representation Nguyen et al. (2010); Nowozin et al. (2016). Using the technique of Noise-Contrastive Estimation (NCE) Gutmann & Hyvärinen (2010), based on comparing target and randomly chosen negative samples, Van den Oord et al. Van den Oord et al. (2018) proposed InfoNCE loss, minimization of which maximizes a mutual information lower bound. An important application of this contrastive learning approach has been extracting high-level representations of different data modalities Chen et al. (2020); Woo et al. (2021); Hu et al. (2021); Koch-Janusz & Ringel (2018). In our work, we extend MINE for transfer entropy estimation.

## 6 CONCLUSION

The problem of reconstructing causal networks from observational data is fundamental in multiple disciplines of science including neuroscience, since it is a prerequisite foundation for the research about structure analysis and behavior control in causal networks. Especially, several countries have recently launched grand brain projects, and one important goal is to map the connectomes (i.e., directed links between neurons) of different species.

We proposed a novel mechanism, *causal attention*, to guide machine learning models to infer causal relationships while focusing on the specific areas where casual effect may emerge. We showed that this mechanism identifies weak causal effects ignored by classical techniques, and helps machine learning models gain insight into the coupling dynamics underlying time series data. Our method needs a small set of samples (i.e., a small number of known causal links), and thus raises an open problem worthy of future pursuit: for large complex systems, how to select the small number of ordered pairs of nodes that offer general pattern for identification of sparse causal effects.

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

## A  DERIVATION

Here we present a derivation showing that the transfer entropy equals the difference between two types of mutual information:

$$TE(X \to Y) = \sum p\left(y_{t+1}, y_t^{(k)}, x_t^{(l)}\right) \log \frac{p\left(y_{t+1} \mid y_t^{(k)}, x_t^{(l)}\right)}{p\left(y_{t+1} \mid y_t^{(k)}\right)}. \tag{12}$$

Applying the conditional Bayes formula $p\left(y \mid x\right) = \frac{p(y,x)}{p(x)}$ on the numerator and denominator in the log term of equation 12:

$$TE(X \to Y) = \sum p\left(y_{t+1}, y_t^{(k)}, x_t^{(l)}\right) \log \frac{\frac{p\left(y_{t+1}, y_t^{(k)}, x_t^{(l)}\right)}{p\left(y_t^{(k)}, x_t^{(l)}\right)}}{\frac{p\left(y_{t+1}, y_t^{(k)}\right)}{p\left(y_t^{(k)}\right)}}. \tag{13}$$

Adding the marginal distribution of time series $\mathbf{Y}$ to the numerator and denominator simultaneously:

$$TE(X \to Y) = \sum p\left(y_{t+1}, y_t^{(k)}, x_t^{(l)}\right) \log \frac{\frac{p\left(y_{t+1}, y_t^{(k)}, x_t^{(l)}\right)}{p(y_r)p\left(y_t^{(k)}, x_t^{(l)}\right)}}{\frac{p\left(y_{t+1}, y_t^{(k)}\right)}{p(y_r)p\left(y_t^{(k)}\right)}} \tag{14}$$

$$= \sum p\left(y_{t+1}, y_t^{(k)}, x_t^{(l)}\right) \log \frac{p\left(y_{t+1}, y_t^{(k)}, x_t^{(l)}\right)}{p\left(y_r\right) p\left(y_t^{(k)}, x_t^{(l)}\right)}$$

$$- \sum p\left(y_{t+1}, y_t^{(k)}\right) \log \frac{p\left(y_{t+1}, y_t^{(k)}\right)}{p\left(y_r\right) p\left(y_t^{(k)}\right)} \tag{15}$$

$$= MI\left(Y_{t+1}, Y_t^{(k)}, X_t^{(l)}\right) - MI\left(Y_{t+1}, Y_t^{(k)}\right). \tag{16}$$

In these expressions, $y_r$ is sampled from time series $\mathbf{Y}$ randomly each time step and independently of the time step $t$.

## B  NEURAL ESTIMATOR FOR TRANSFER ENTROPY

### B.1  CONSISTENCY

**Definition.** A neural estimator $\widehat{S}(X,Y)_n$ which uses $n$ samples from the data distribution to estimate a statistic $S(X,Y)$ on variables $X, Y$ is *strongly consistent* if for any $\epsilon > 0$, there exists a positive integer $N$ and a choice of neural network such that:

$$\forall n \geq N, \quad \mid S(X,Y) - S(X,Y)_n \mid \leq \epsilon, \text{almost everywhere (a.e.)} \tag{17}$$

The Mutual Information Neural Estimator (MINE) depends on a choice of a neural network and the number of samples $n$ from the data distribution Belghazi et al. (2018). Let $f_\theta$ be the family of functions parameterized by the neural network with parameters $\theta \in \Theta$. MINE is defined as:

$$\widehat{MI}(X,Y)_n = \sup_{\theta \in \Theta} E_{P_{XY}^{(n)}} \left[f_\theta\right] - \log \left(E_{P_X^{(n)} \otimes P_Y^{(n)}} \left[e^{f_\theta}\right]\right). \tag{18}$$

**Theorem 1** Belghazi et al. (2018). MINE is strongly consistent.

The Transfer Entropy Neural Estimator (TENE) consists of two independent MINE and depends on choice of neural network and sample number $n$. TENE is defined as:

$$\widehat{TE}(X \to Y)_n = \widehat{MI}\left(Y_{t+1}, \left(Y_t^{(k)}, X_t^{(l)}\right)\right)_n - \widehat{MI}\left(Y_{t+1}, Y_t^{(k)}\right)_n. \tag{19}$$

We use $MI^{[1]}$, $MI^{[2]}$, $\widehat{MI}_n^{[1]}$, $\widehat{MI}_n^{[2]}$ as abbreviations of $MI\left(Y_{t+1}, \left(Y_t^{(k)}, X_t^{(l)}\right)\right)$ and $MI\left(Y_{t+1}, Y_t^{(k)}\right)$, $\widehat{MI}\left(Y_{t+1}, \left(Y_t^{(k)}, X_t^{(l)}\right)\right)_n$ and $\widehat{MI}\left(Y_{t+1}, Y_t^{(k)}\right)_n$ respectively.

We will prove the following:

**Theorem 2.** TENE is strongly consistent.

**Proof.** Let $\epsilon > 0$. By Theorem 1, we can choose neural networks and integers $N_1, N_2$ and such that

$$\forall n \geq N_1, \quad \left|MI^{[1]} - \widehat{MI}_n^{[1]}\right| \leq \epsilon/2, \text{a.e.} \tag{20}$$

$$\forall n \geq N_2, \quad \left|MI^{[2]} - \widehat{MI}_n^{[2]}\right| \leq \epsilon/2, \text{a.e.} \tag{21}$$

Letting $N = \max\{N_1, N_2\}$, for $n \geq N$ and for some neural network we have, a.e.,

$$\forall n \geq N, \left|TE(X \to Y) - \widehat{TE}(X \to Y)_n\right| = \left|(MI^{[1]} - MI^{[2]}) - (\widehat{MI}_n^{[1]} - \widehat{MI}_n^{[2]})\right| \tag{22}$$

$$= \left|(MI^{[1]} - \widehat{MI}_n^{[1]}) - (MI^{[2]} - \widehat{MI}_n^{[2]})\right| \tag{23}$$

$$\leq \left|(MI^{[1]} - \widehat{MI}_n^{[1]})\right| + \left|(MI^{[2]} - \widehat{MI}_n^{[2]})\right| \tag{24}$$

$$\leq \epsilon/2 + \epsilon/2 = \epsilon. \tag{25}$$

The proof is complete.

## B.2 VARIATION OF BIAS VARY WITH DIMENSION AND NOISE

We examine the performance of TENE for the considered class of neural networks on linear dynamic system, consisting of variables $X$ and $Y$ defined as:

$$x_{t+1} = \alpha x_t + \varepsilon_x \tag{26}$$

$$y_{t+1} = \beta y_t + g_c x_t + \varepsilon_y \tag{27}$$

We set $\alpha = \beta = 0.5$ and $\varepsilon_x = \varepsilon_x \sim N(0, \sigma^2)$. The true values of transfer entropy $TE(X \to Y)$ in this simple coupled system can be determined analytically Kaiser & Schreiber (2002). We can increase the dimension of the system by considering multiple independent copies of variables $X$ and $Y$, in which case the mutual information and transfer entropy scale linearly with the dimension of the system. For each considered dimension, standard deviation $\sigma$, and coupling strength $g_c$ in an interval from -0.4 to 0.4, we generate a time series of length 50,000. We also consider an alternative non-parametric estimator of mutual information, the Kraskov estimator Kraskov et al. (2004) with $k = 5$ nearest neighbours. In Fig 5 we compare the results of MINE with the analytic formula and the Kraskov estimator. MINE shows marked improvement over the Kraskov estimator, especially when variables are high-dimensional. Comparing Fig 5(a,b) or (c,d) shows that the amplitude of the driving Gaussian noise has little influence on estimates. Interestingly, as coupling strength $g_c$ grows small, i.e., as $X$ and $Y$ become more independent, the Kraskov estimator can suggest a negative value of the mutual information, i.e., we estimate that $MI_n(Y_{t+1}, Y_t, X_t) < MI_n(Y_{t+1}, Y_t)$. We deduce that irrelevant information about the nearly independent variable $X_t$ interferes with the estimation of the mutual information by the Kraskov estimator.

## C ALGORITHM

Details on the implementation of causal attention mechanism are provided in Algorithm 1.

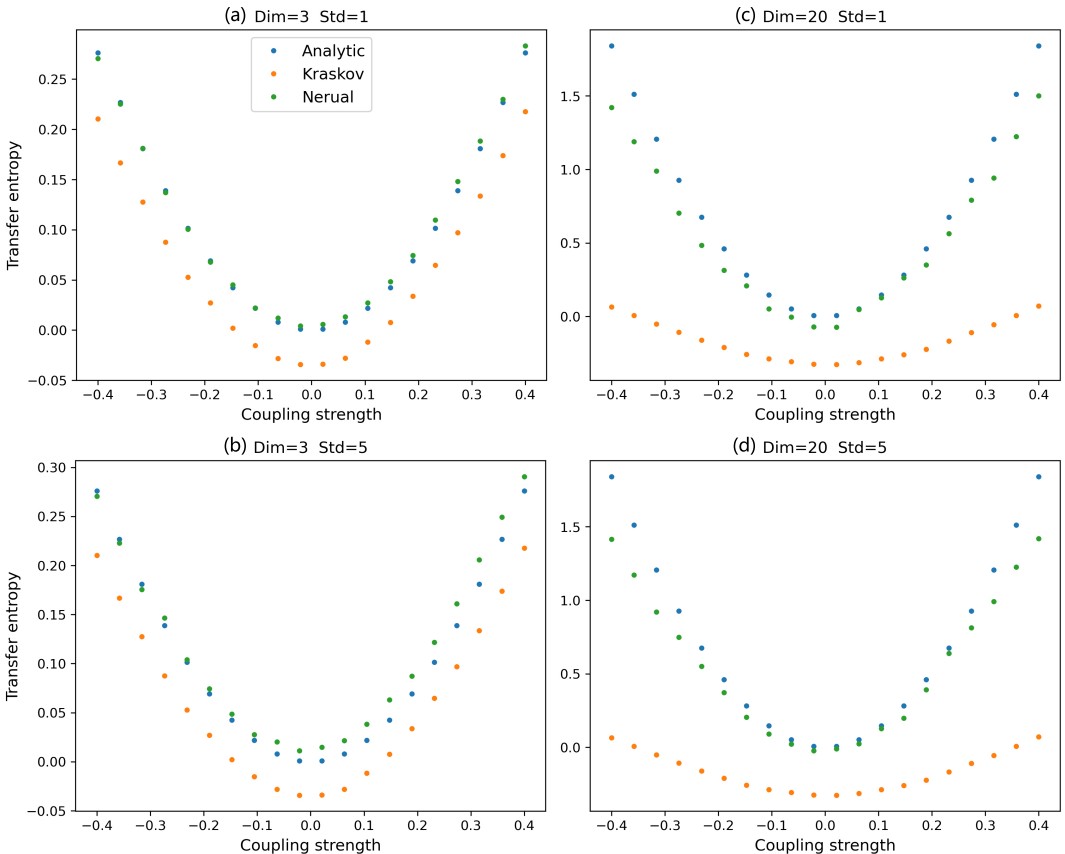

Figure 5: True and estimated transfer entropy versus coupling strength $g_c$. The dimension and standard deviation (std) $\sigma$ of system noise is indicated in the titles of subplots.

---

**Algorithm 1** Causal Inference by Causal Attention

**Input:** Small samples $S$, part with causality $\bar{S} \subset S$, segment length for training $\bar{L} \ll L$

1: $\theta, \phi, \alpha, \eta \leftarrow$ initialize network parameters
2: **repeat until** $(\mathcal{L}_1 - \mathcal{L}_2)$ is stable in optimal loss interval
3:     **for** all samples $\bar{S}$ **do**
4:         Choose $t_s \in \left[1, L - \bar{L}\right]$ randomly
5:         Take segment $s = \left\{(x_t, y_t)\right\}_{t=t_s}^{t_s + \bar{L}}$
6:         Produce joint samples in Eq. 8
7:         $\left\{\left(y_{t+1}, y_t^{(k)}, x_t^{(l)}\right)\right\}_{t=t_s}^{t_s + \bar{L}}$, etc.
8:     **end for**
9:     Assign causal attention $A$
10:    Compute $\mathcal{L}_1, \mathcal{L}_2$ on $\bar{S}$
11:    Update parameters    $\theta, \phi \leftarrow \theta + \nabla_\theta \mathcal{L}_1, \phi + \nabla_\phi \mathcal{L}_2$
12:    Recompute $\mathcal{L}_1, \mathcal{L}_2$ on $\hat{S}$
13:    Update parameters    $\alpha \leftarrow \alpha + \nabla_\alpha \left(\mathcal{L}_1 - \mathcal{L}_2\right)$
14: **repeat until** $\mathcal{L}_3$ convergence
15:    Compute $\mathcal{L}_3$ on $S$
16:    Update parameters    $\eta \leftarrow \eta - \nabla_\eta \mathcal{L}_3$

---

## D  MODEL BRAIN DYNAMICS

Here we give detailed information about five neuronal dynamics applied to modeling membrane potential and relevant quantities in biological connectomes. We input to each causal discovery algorithm the coordinate corresponding to the neuron membrane voltage potential, because this variable is most likely to be experimentally accessible.

### D.1  HINDMARSH-ROSE DYNAMICS

The spikes of activity in neurons are considered an important part of the brain's information processing Borges et al. (2018); Rabinovich et al. (2006). Hindmarsh and Rose Hindmarsh & Rose (1984) (HR) proposed a phenomenological neuron model that is a simplification of the Hodgkin-Huxley model Hodgkin & Huxley (1952). The HR model is described by

$$\dot{p} = q - ap^3 + bp^2 - n + I_{\text{ext}}$$
$$\dot{q} = c - dp^2 - q$$
$$\dot{n} = r\left[s\left(p - p_0\right) - n\right]$$

where $p(t)$ is the action potential of the membrane, $q(t)$ is related to the fast current and $n(t)$ is associated with the slow current. Presynaptic neurons with an action potential $p_j$ coupled by chemical synapses to neurons $i$ modifying its action potential $p_i$ according to

$$\dot{p}_i = q_i - ap_i^3 + bp_i^2 - n + I_{\text{ext}} + \Gamma$$
$$\Gamma = g_c\left(V_{\text{syn}} - p_i\right)\textstyle\sum_{j=1}^{N}\frac{B_{ij}}{1 + \exp(-\lambda\left(p_j - \Theta_{\text{syn}}\right))}$$

where $i, j = 1, \ldots, N$, $N$ is the number of neurons, $g_c$ is the chemical coupling strength and $B_{ij}$ describes neurons' chemical connections. The chemical synapse function is modeled by the above sigmoidal function, with $\Theta_{\text{syn}} = 1.0$. We use parameters $a = 1, b = 3, c = 1, u = 5, s = 4, r = 0.005, p_0 = -1.60$, coupling strength $g_c = 0.1, V_{\text{syn}} = 2, \lambda = 10$, and external current $I_{\text{ext}} = 3.24$, for which HR neurons exhibits a chaotic burst behavior.

### D.2  MORRIS–LECAR DYNAMICS

Morris and Lecar  Morris & Lecar (1981) suggested a simple two variable model to describe oscillations in a barnacle's giant muscle fiber. The Morris–Lecar model has became quite popular in computational neuroscience community due to its biophysically meaningful and measurable parameters, which consist of a membrane potential $u$ receiving an instantaneously activated Ca current and a more slowly activated K current $n$ evolving according to:

$$C\dot{V} = I - g_L\left(V - V_L\right) - g_{\text{Ca}}m_\infty(V)\left(V - V_{\text{Ca}}\right) - g_K n\left(V - V_K\right) + \Gamma(V)$$
$$\dot{n} = \lambda(V)\left(n_\infty(V) - n\right)$$

where

$$m_\infty(V) = \frac{1}{2}\left\{1 + \tanh\left[\frac{(V - V_1)}{V_2}\right]\right\}$$
$$n_\infty(V) = \frac{1}{2}\left\{1 + \tanh\left[\frac{(V - V_3)}{V_4}\right]\right\}$$
$$\lambda(V) = \bar{\lambda}\cosh\left[\frac{(V - V_3)}{(2V_4)}\right]$$

with the coupling term

$$\Gamma(V_i) = g_c\textstyle\sum_{j=1}^{N} B_{ij}\left(n_j - n_i\right),$$

with parameters $C = 20\mu\text{F}/\text{cm}^2, g_L = 2\text{mmho}/\text{cm}^2, V_L = -50\text{mV}, g_{\text{Ca}} = 4\text{mmho}/\text{cm}^2, V_{\text{Ca}} = 100\text{mV}, g_K = 8\text{mmho}/\text{cm}^2, V_K = -70\text{mV}, V_1 = 0\text{mV}, V_2 = 15\text{mV}, V_3 = 10\text{mV}, V_4 = 10\text{mV}, \bar{\lambda} = 0.1 \text{ s}^{-1}$, and applied current $I = 34\mu\text{A}/\text{cm}^2$.

### D.3 IZHIKEVICH DYNAMICS

Izhikevich dynamics reproduce spiking and bursting behavior of known types of cortical neurons, and combine the biological plausibility of Hodgkin–Huxley-type dynamics and the computational efficiency of integrate-and-fire neurons Izhikevich (2003). The equations governing Izhikevich spike dynamics are:

$$\dot{v} = 0.04v^2 + 5v + 140 - u + I + g_c \sum B_{ij} u_j$$
$$\dot{u} = a(bv - u)$$

with the auxiliary after-spike resetting

$$\text{if } v \geq +30\text{mV}, \quad \text{then } \begin{cases} v \leftarrow c \\ u \leftarrow u + d \end{cases}.$$

Here, variable $v$ represents the membrane potential of the neuron and $u$ represents a membrane recovery variable, which accounts for the activation of $K^+$ ionic currents and inactivation of $\text{Na}^+$ ionic currents, and it provides negative feedback to $v$. Here, we use the parameters $a = 0.2, b = 2, c = -56, d = -16, I = -99$. After the spike reaches its apex ($+30\text{mV}$), the membrane voltage and the recovery variable are reset. If $v$ skips over 30 , then it first is reset to 30 , and then to $c$ so that all spikes have equal magnitudes.

### D.4 RULKOV DYNAMICS

The Rulkov model is a map-based neuron model with a surprising abundance of features, such as periodic and chaotic spiking, and bursting. The Rulkov map is an abstract mathematical model, although it shares some specific features with others neuron models closer to experimental observations. We use synthetic time series where each neuron is simulated using the Rulkov model Eroglu et al. (2020), which has two variables, $u$ and $w$, evolving at different timescales as described by $\boldsymbol{x}(t + 1) = (u(t + 1), v(t + 1)) = \boldsymbol{F}(\boldsymbol{x}(t)) = (F_1(u(t), w(t)), F_2(u(t), w(t)))$, with

$$F_1(u, w) = \frac{\beta}{1 + u^2} + w + \Gamma(u) \quad \text{and} \quad F_2(u, w) = w - \nu u - \sigma.$$

The two variables reflect the two important time scales of a neuron model. The variable $u$ represents the fast dynamics of the system and usually models the membrane voltage of the neuron, whereas $w$ is the slow variable and represents the variations of the ionic recovery currents. Different combinations of parameters $\sigma$ and $\beta$ give rise to different dynamical states of the neuron, such as resting, tonic spiking, and chaotic bursts. As for the coupling, we consider chemical synaptic coupling, that is, $\boldsymbol{H}(\boldsymbol{x}_i, \boldsymbol{x}_j) = (h(u_i, u_j), 0)$ with $h(u_i, u_j) = (u_i - V_s)\Gamma(u_j)$, where

$$\Gamma(u_j) = \frac{1}{1 + \exp\{\lambda(u_j - \Theta_s)\}}.$$

and electrical synaptic coupling, $\boldsymbol{H}(\boldsymbol{x}_i, \boldsymbol{x}_j) = (h(u_i, u_j), 0)$, with $h(u_i, u_j) = u_j - u_i$. In the chemical coupling, $V_s$ is a parameter called the reverse potential. Here, we use the parameters with $\beta = 4.4, \sigma = \nu = 0.001,, V_s = 20, \Theta_s = -0.25$ and $\lambda = 10$.

### D.5 FITZHUGH-NAGUMO DYNAMICS

A FitzHugh-Nagumo neuron comprises a two-dimensional system of smooth ODEs, so cannot exhibit autonomous chaotic dynamics and bursting. Adding noise allows for stochastic bursting FitzHugh (1961). The equations governing the FitzHugh-Nagumo neuronal network dynamics are

$$\dot{v} = a + bv + cv^2 + dv^3 - u + \Gamma$$
$$\dot{u} = \varepsilon(ev - u)$$

with the coupling term

$$\Gamma(v_i) = -g_c \sum_{j=1}^{N} B_{ij}(v_j - v_i).$$

The FitzHugh-Nagumo dynamics capture the firing behaviors of neurons with two components. The first component $v$ represents the membrane potential, which contains self- and interaction dynamics,

and the second component $u$ represents a recovery variable. To simulate the shape of each spike, the time step in the model must be relatively small, e.g., $\tau = 0.25$ ms. Here, we use the parameters $a = 0.28, b = 1, c = 0, d = -1, \varepsilon = 0.04, e = 12.5$. Moreover, the parameters in the FitzHugh–Nagumo model can be tuned so that the model describes spiking dynamics of many resonator neurons.

### D.6 Time series generation

To obtain the time series from above neural dynamics, we use Runge-Kutta method with variable-step to solve the ordinary differential equation of Hindmarsh-Rose and Morris–Lecar dynamics with sample interval $\tau = 0.1$. Izhikevich dynamics are solved by the Euler formula with time step $h = 0.05$. For the Rulkov map we consider a unit sample interval. The total time step of time series $T = 50,000$ in both synthetic and real networks.

## E  Real brain connectomes information

### E.1 Cat Connectome

The cat connectivity dataset comprises a description of cortical connections in the cat brain Scannell et al. (1995), a connectivity set resulting from a comprehensive literature search of anatomical tracing studies in the cat cortex. Detailed information on the delineated regions, including information on the used parcellation scheme, abbreviations and possible overlap with other parcellation schemes, as well as information on the physiological characteristics of these regions, is given in the appendix of the original study Ref. Scannell et al. (1995). The connectivity dataset incorporates data of one hemisphere, including 65 regions and 1139 interregional macroscopic axonal projections de Reus & van den Heuvel (2013).

### E.2 Macaque Connectome

The macaque connectivity data set used in this study comprises anatomical data from 410 tract tracing studies collated in the online neuroinformatics data base CoCoMac (http://cocomac.org), first analyzed and made publicly available in Ref. Modha & Singh (2010). In the present study they focused primarily on an analysis of the connectivity among regions of the cerebral cortex. The cortical connection matrix was extracted from the primary connection data by removing all subcortical (thalamus, basal ganglia, brainstem) regions. In addition, regions that did not maintain at least one incoming and one outgoing connection were also removed to ensure that the network was strongly connected. The remaining connection data set used in this study consisted of 242 regions and 4090 directed projections represented in binary format (connection present = 1, connection absent = 0) Harriger et al. (2012).

### E.3 Mouse Connectome

The Allen Mouse Brain Connectivity Atlas uses enhanced green fluorescent protein (EGFP)-expressing adeno-associated viral vectors to trace axonal projections from defined regions and cell types, and high-throughput serial two-photon tomography to image the EGFP-labelled axons throughout the brain. This systematic and standardized approach allows spatial registration of individual experiments into a common three dimensional (3D) reference space, resulting in a whole-brain connectivity matrix. The Allen Mouse Brain Connectivity Atlas is a freely available, foundational resource for structural and functional investigations into the neural circuits that support behavioural and cognitive processes in health and disease Oh et al. (2014).

### E.4 Worm Connectome

All the chemical and gap junction synapses, the connectome, in the posterior nervous system of the C. elegans adult male are identified by serial section electron microscopy Jarrell et al. (2012). The feasibility of comprehensive synapse-level nervous system reconstruction by this method was a primary reason for the initial selection of C. elegans as an experimental model. They developed a PC-based software platform to facilitate assembly of a connectome from electron micrographic

images. The connectome is of a single adult animal and was produced from a series of 5000 serial thin sections of 70 to 90 nm encompassing the posterior half of the body.

### E.5 RAT CONNECTOME

Because resliceable 3D brain models for relating systematically and topographically different parcellation schemes are still in the first phases of development, it is necessary to rely on qualitative comparisons between regions and tracts that are either inserted directly by neuroanatomists or trained annotators, or are extracted or inferred by collators from the available literature. To address these challenges, Ref. Bota et al. (2012) developed a publicly available neuroinformatics system, the Brain Architecture Knowledge Management System, including an exemplar for constructing interrelated connectomes at different levels of the mammalian central nervous system organization, and presented the latest version of the BAMS rat macroconnectome.

Information about the above datasets is summarized in Table 4.

Table 4: Statistical information of six real networks: dataset name, type of network, number of nodes, number of edges, mean degree $\langle k \rangle$, and data acquisition method.

| Dataset | Region | #Nodes | #Edges | Mean degree | Sensor |
|---------|--------|--------|--------|-------------|--------|
| Cat | Brain | 65 | 1139 | 17.5 | Tract tracing studies |
| Macaque | Brain | 242 | 4090 | 16.9 | Tract tracing studies |
| Mouse | Cerebral Cortex | 195 | 214 | 1.1 | Electron microscopy |
| Worm | Neural | 272 | 4451 | 16.4 | Electron Microscopy |
| Rat | Brain | 503 | 30088 | 59.8 | Neuroanatomical experiments |

## F ADDITIONAL EXPERIMENTS

### F.1 PERFORMANCE OF TRANSFER ENTROPY ON REAL CAUSAL NETWORKS

Experiment Results on Macaque/C.elegans/Rat connectome are provided in Table 5/Table 6/Table 7 as the supplement of main text Table 3. Classic methods have limited performance across various neural dynamics unfolding on real causal networks especially in a noisy environment: we add Gaussian measurement noise with mean zero and standard deviation $10\%$ that of the original time series. As we discuss in main text Sec. 3.1, sparse causal effects are easily masked in metric of iconic transfer entropy when noise causes estimated probability densities to deviate even slightly from the true distributions.

Table 5: Performance comparison on Macaque connectome. The sample number in train/validation/test set is 50/50/500.

| | Hindmarsh-Rose | | Morris-Lecar | | Izhikevich | | Rulkov | | FitzHugh-Nagumo | |
|---|---|---|---|---|---|---|---|---|---|---|
| **Granger** | 0.50±0.01 | 0.50±0.01 | 0.48±0.02 | 0.48±0.01 | 0.50±0.02 | 0.50±0.01 | 0.54±0.01 | 0.54±0.01 | 0.54±0.01 | 0.54±0.01 |
| **TE** | 0.44±0.01 | 0.49±0.01 | 0.53±0.05 | 0.52±0.05 | 0.54±0.03 | 0.51±0.02 | 0.48±0.07 | 0.49±0.04 | 0.43±0.02 | 0.45±0.01 |
| **CCM** | 0.44±0.02 | 0.44±0.02 | 0.49±0.01 | 0.49±0.01 | 0.51±0.01 | 0.51±0.01 | 0.55±0.01 | 0.55±0.01 | 0.56±0.01 | 0.56±0.01 |
| **Latent CCM** | 0.47±0.01 | 0.47±0.01 | 0.51±0.02 | 0.51±0.01 | 0.51±0.01 | 0.51±0.02 | 0.53±0.02 | 0.53±0.02 | 0.53±0.01 | 0.53±0.01 |
| **PCMCI** | 0.47±0.01 | 0.47±0.01 | 0.51±0.01 | 0.51±0.01 | 0.50±0.02 | 0.50±0.01 | 0.50±0.01 | 0.50±0.01 | 0.52±0.01 | 0.52±0.01 |
| **PCMCI⁺** | 0.47±0.02 | 0.47±0.01 | 0.48±0.01 | 0.48±0.01 | 0.51±0.02 | 0.51±0.01 | 0.52±0.02 | 0.52±0.01 | 0.52±0.01 | 0.52±0.01 |
| **TA** | 0.68±0.03 | 0.51±0.03 | 0.56±0.01 | 0.55±0.01 | 0.52±0.04 | 0.50±0.02 | 0.51±0.02 | 0.51±0.03 | 0.54±0.01 | 0.52±0.01 |
| **ATE** | **0.71**±0.04 | **0.65**±0.03 | **0.60**±0.02 | **0.59**±0.01 | **0.66**±0.03 | **0.64**±0.03 | **0.59**±0.03 | **0.57**±0.03 | **0.63**±0.02 | **0.59**±0.02 |

Table 6: Performance comparison on C.elegans connectome. The sample number in train/validation/test set is 50/50/500.

| | Hindmarsh-Rose | | Morris-Lecar | | Izhikevich | | Rulkov | | FitzHugh-Nagumo | |
|---|---|---|---|---|---|---|---|---|---|---|
| Granger | 0.50±0.01 | 0.50±0.01 | 0.56±0.01 | 0.56±0.01 | 0.64±0.01 | 0.64±0.02 | 0.71±0.01 | 0.71±0.01 | 0.64±0.02 | 0.64±0.01 |
| TE | 0.74±0.01 | 0.73±0.02 | 0.61±0.01 | 0.58±0.02 | 0.55±0.01 | 0.56±0.02 | 0.80±0.01 | 0.80±0.01 | 0.52±0.01 | 0.48±0.01 |
| CCM | 0.79±0.02 | 0.79±0.02 | 0.54±0.02 | 0.54±0.02 | 0.62±0.02 | 0.62±0.02 | 0.67±0.02 | 0.67±0.02 | 0.74±0.01 | 0.74±0.01 |
| Latent CCM | 0.78±0.01 | 0.78±0.01 | 0.52±0.01 | 0.52±0.01 | 0.55±0.04 | 0.55±0.03 | 0.53±0.04 | 0.53±0.03 | 0.68±0.01 | 0.68±0.01 |
| PCMCI | 0.53±0.01 | 0.53±0.01 | 0.50±0.02 | 0.50±0.01 | 0.53±0.02 | 0.53±0.01 | 0.66±0.01 | 0.66±0.01 | 0.57±0.01 | 0.59±0.01 |
| PCMCI⁺ | 0.53±0.02 | 0.53±0.01 | 0.52±0.01 | 0.52±0.01 | 0.56±0.01 | 0.56±0.01 | 0.69±0.02 | 0.69±0.02 | 0.63±0.01 | 0.63±0.01 |
| TA | 0.92±0.01 | 0.91±0.01 | 0.67±0.03 | 0.65±0.03 | 0.52±0.01 | 0.54±0.01 | 0.56±0.01 | 0.54±0.01 | **0.85**±0.01 | **0.84**±0.01 |
| ATE | **0.93**±0.01 | **0.92**±0.01 | **0.81**±0.01 | **0.76**±0.01 | **0.76**±0.01 | **0.71**±0.01 | **0.97**±0.01 | **0.97**±0.01 | 0.78±0.01 | 0.72±0.02 |

Table 7: Performance comparison on Rat connectome. The sample number in train/validation/test set is 100/100/500.

| | Hindmarsh-Rose | | Morris-Lecar | | Izhikevich | | Rulkov | | FitzHugh-Nagumo | |
|---|---|---|---|---|---|---|---|---|---|---|
| Granger | 0.50±0.01 | 0.50±0.01 | 0.43±0.02 | 0.43±0.01 | 0.50±0.01 | 0.50±0.01 | 0.46±0.01 | 0.46±0.01 | 0.52±0.02 | 0.52±0.01 |
| TE | 0.57±0.02 | 0.62±0.01 | 0.33±0.02 | 0.41±0.01 | 0.48±0.09 | 0.54±0.08 | 0.37±0.01 | 0.43±0.01 | 0.41±0.03 | 0.47±0.03 |
| CCM | 0.63±0.02 | 0.63±0.02 | 0.48±0.02 | 0.48±0.02 | 0.54±0.05 | 0.54±0.05 | 0.50±0.01 | 0.50±0.01 | 0.52±0.03 | 0.52±0.02 |
| Latent CCM | 0.63±0.02 | 0.63±0.02 | 0.40±0.01 | 0.40±0.01 | 0.55±0.01 | 0.55±0.02 | 0.53±0.02 | 0.53±0.01 | 0.51±0.03 | 0.51±0.02 |
| PCMCI | 0.51±0.01 | 0.51±0.01 | 0.44±0.02 | 0.44±0.01 | 0.53±0.01 | 0.53±0.01 | 0.46±0.01 | 0.46±0.01 | 0.49±0.02 | 0.49±0.01 |
| PCMCI⁺ | 0.51±0.02 | 0.51±0.01 | 0.42±0.02 | 0.42±0.01 | 0.55±0.01 | 0.55±0.01 | 0.44±0.02 | 0.44±0.01 | 0.49±0.01 | 0.49±0.01 |
| TA | 0.82±0.03 | 0.78±0.02 | 0.70±0.01 | 0.67±0.01 | 0.62±0.02 | 0.58±0.03 | 0.73±0.02 | 0.74±0.01 | 0.74±0.04 | 0.74±0.02 |
| ATE | **0.83**±0.03 | **0.83**±0.03 | **0.78**±0.01 | **0.78**±0.01 | **0.65**±0.04 | **0.64**±0.03 | **0.83**±0.01 | **0.79**±0.01 | **0.84**±0.02 | **0.81**±0.01 |

## F.2 DATA EFFICIENCY OF CAUSAL ATTENTION MECHANISM AGAINST THE LENGTH OF TIME SERIES

Taking Cat and Mouse connectome as examples, we show the results of our method trained on the time series data with different lengths, i.e. total time steps, in Figure 6. The size of training and test sets is the same as the scheme in the main text Table 4. Overall, the AUROC/AUPRC scores tend to get higher as the length increases, but this tendency is not significant. It indicates that the causal attention mechanism extracts sufficient causal features for causal inference even over short time series.

## F.3 ROBUSTNESS OF CAUSAL ATTENTION MECHANISM AGAINST THE INTENSITY OF NOISE

We show the results of our method trained on time series data added different intensities of noise (measurement noise rather than intrinsic noise of dynamics) in Figure 7. Except for dynamics of Izh and FHN on Cat connectome, the AUROC/AUPRC scores are stable within the range 2%-10% of standard deviation. It implies that causal attention mechanism is robust by the sample noise level.

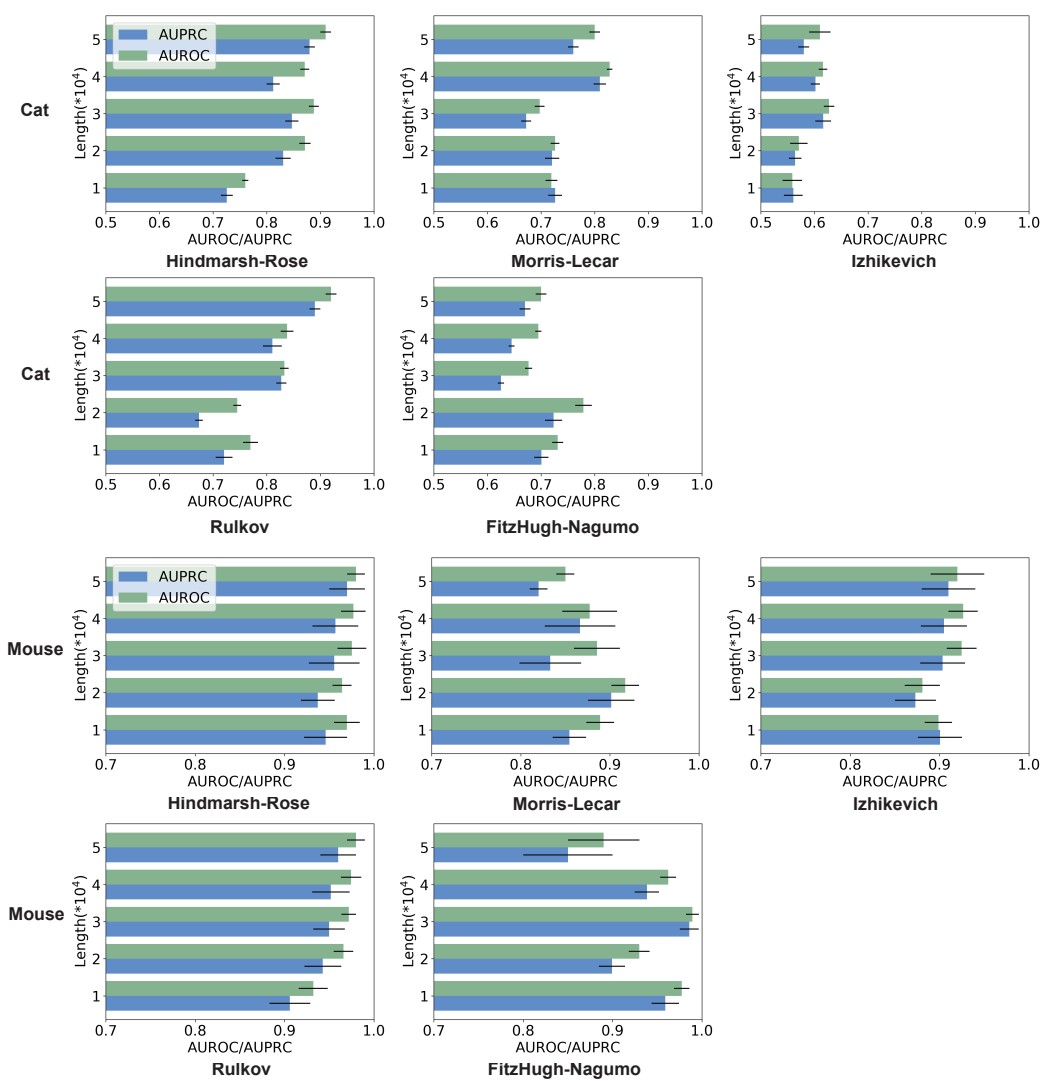

Figure 6: AUROC/AUPRC scores of the *causal attention* mechanism trained on time series data of different lengths. The blue bar is AUPRC, the green bar is AUROC. The x-axis indicates the scores and y-axis represents the total time step of time series.

### F.4 HOW MUCH DOES CAUSAL ATTENTION MECHANISM BRING THE DISTANCE BETWEEN DISTRIBUTIONS OF TRAINING AND TEST SET CLOSER TOGETHER

The causal attention mechanism helps the classifier reveal the generation processing underlying the data, i.e., coupling-drive in problem of causal inference and alleviate the dilemma of distribution shift in scene of small samples. The causal attention mechanism refines the content of samples (critical regions) and thus reduces the distribution dimension of the entire dataset. To quantify this shortened distance, we ask, how many additional training samples does traditional machine learning need to achieve the same level of generalization as our method? Taking the Cat Connectome as example, we train the classifier with traditional attention mechanism by gradually expanding the size of training set, and provide the results in Figure8. The green horizontal lines represent the AUROC value of our method using ten ordered pairs with causality ($0.8\%$ of edges in Cat Connectome) while, to achieve the same performance, the traditional classifier needs approximately $10\%/15\%/20\%/40\%$ samples for HR/Izh/Morris/Rulkov dynamics respectively (the blue lines cross the green lines). It also indicates that our method provides significant saving in labels collection, which is significant given that the procedure for checking connections in organisms is cumbersome in practice.

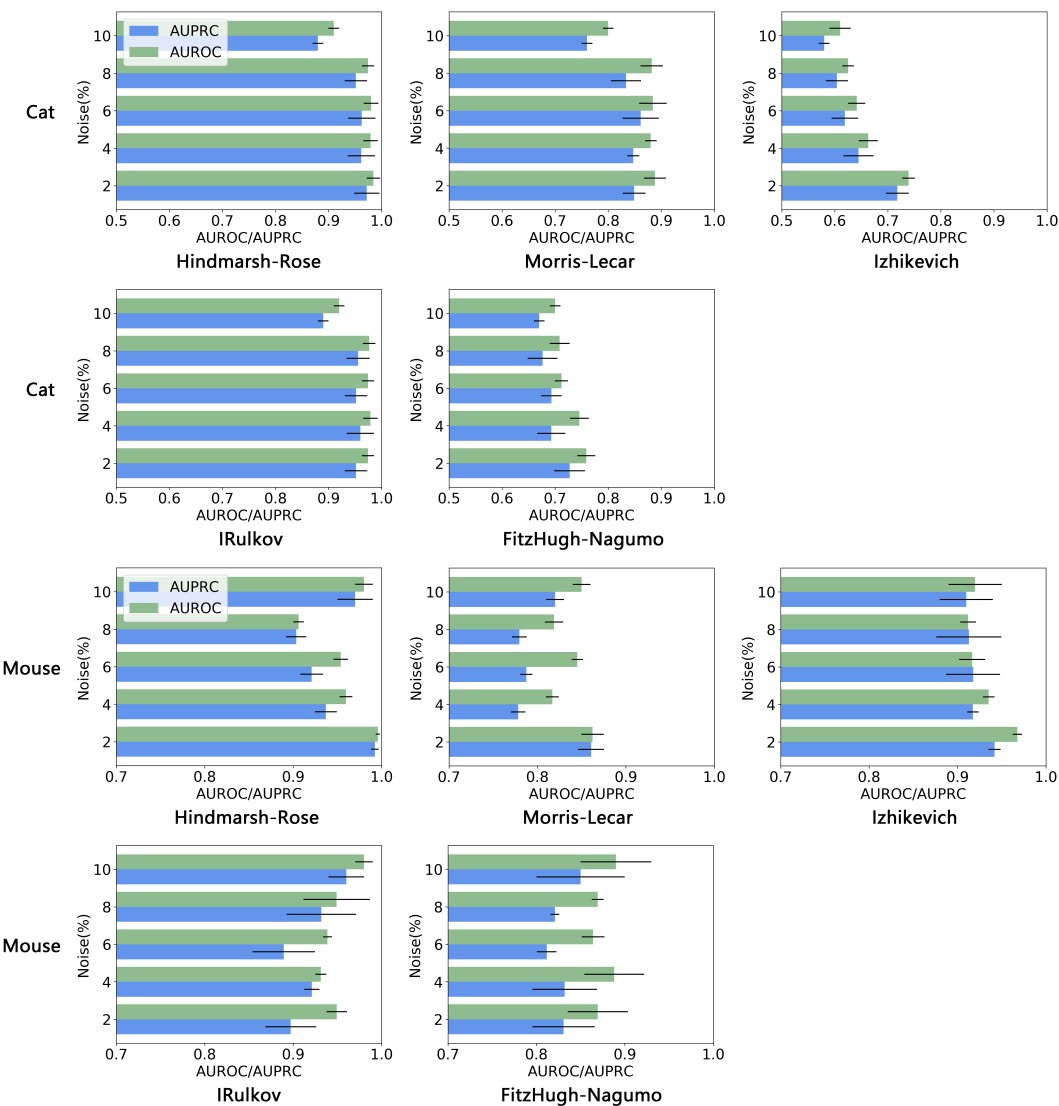

Figure 7: AUROC/AUPRC scores of the *causal attention* mechanism trained on time series data added different intensity of noise. The blue bar is AUPRC, the green bar is AUROC, and the y-axis indicates the percentage of standard deviation of Gaussian measurement noise in the original time series.

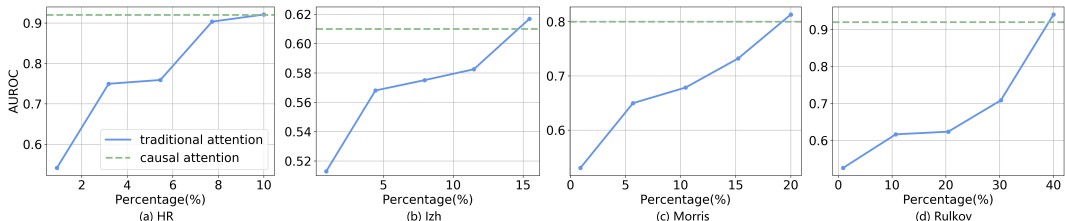

Figure 8: Size of training set that the traditional classifier requires to achieve same level of generalization as our method. The x-coordinate indicates the percentage of the ordered pairs with causality in training set to the total edges in causal network. Examples of dynamical model: (a) HR; (b) Izh; (c) Morris; (d) Rulkov.

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
