# OpenReview forum: "Causal Attention to Exploit Transient Emergence of Causal Effect"
_ICLR.cc/2023/Conference — Submitted to ICLR 2023_

### Official Review · Reviewer_Ube5 · 2022-10-23

**Confidence:** 4
**Clarity, Quality, Novelty And Reproducibility:** Good written paper.
**Correctness:** 3
**Technical Novelty And Significance:** 3
**Empirical Novelty And Significance:** 3
**Recommendation:** 8

**Strength And Weaknesses:**

My recommendation to the authors is to present the limitations of this type of methodologies right after the contributions in the introduction. So, a new reader can quickly understand where and how this methodology can be applied. One of the limitations is the degradation of performance as the degree increases. It’s certainly problem dependent but it can be useful to know in what problem one can potentially apply this methodology. Another limitation is for dynamical systems with strong perturbations when, for example, synchronization phenomena dominate the system dynamics. The authors also show in the appendix that the method is quite sensitive to noise, some something should be written in this regard. It also appears to me that there’s quite a lot of effort setting up the right balance between the learning rates and the attention variable such that the cross-entropy loss leads to good generalization. Having guardrails on how to set the balance between the a-values and the cross-entropy loss would facilitate extending this method to other contexts.

**Summary Of The Paper:**

The authors propose a novel mechanism of attention to train neural networks to identify the coupling parameters of complex dynamical system under small perturbations. The problem belongs to the space of dynamical systems with strongly dissipative elements with small coupling strength. The idea is inspired by Schriber, 2000 to use transfer entropy to measure uncertainty in the causal relationships between variables. The results shown large improvements respect to other methods. And the attention variables shown in Fig. 3 appear to capture well the state-space volumes where external perturbations from other neurons/elements are expected to have larger impact. When the neurons are near the resting state perturbations are quickly erased, while exiting the saddles through the unstable separatrix amplifies the perturbation effect and the signal-to-noise ratio is expected to be more favorable. So, the method makes quite a lot of sense to me.

My recommendation to the authors is to present the limitations of this type of methodologies right after the contributions in the introduction. So, a new reader can quickly understand where and how this methodology can be applied. One of the limitations is the degradation of performance as the degree increases. It’s certainly problem dependent but it can be useful to know in what problem one can potentially apply this methodology. Another limitation is for dynamical systems with strong perturbations when, for example, synchronization phenomena dominate the system dynamics. The authors also show in the appendix that the method is quite sensitive to noise, some something should be written in this regard. It also appears to me that there’s quite a lot of effort setting up the right balance between the learning rates and the attention variable such that the cross-entropy loss leads to good generalization. Having guardrails on how to set the balance between the a-values and the cross-entropy loss would facilitate extending this method to other contexts.

Another more specific comment that I wish the authors could have addressed is the intended consequences of the mechanism of attention in the context of stable and unstable (saddles) fixed points in dissipative dynamical systems. The fixed points and their separatrices form the skeleton of trajectories and the saddles are the escape routes away of a resting state. Moreover, the neuron models the authors can have bistability and another neuron can flip the child one from resting state to the limit cycle, for example. Once in the limit cycle there will be very tiny modifications on the frequency of oscillations, but the pulse that switched the neuron away from the resting state is short lived. So, if one understands the bifurcations of the individual neuron dynamics, then we can infer where in the state-space external perturbations will lead to the largest information gains and, perhaps, we do not need the attention mechanism.

Minor comments:
In the appendix, injecting noise in a dissipative dynamical system is not equivalent as adding 10% noise to the time series. The noise gets integrated through any SDE solver. It leads to compression of the noise in the highly dissipative regimes and gets amplified nearby the unstable saddles. So, it does work in a similar way as the attention mechanism (or works against it), because it can also be used to identify the state-space volumes where the entropy gets amplified.


Intro: Replace the rhetorical question: “Why are information from parent ..?” by a direct sentence.

ATE is commonly used as average treatment effect in classic causal effect estimations. So, it may lead to confusion when people from other fields read this paper in the context of causality.





**Summary Of The Review:**

Novel contribution well supported by plenty of experiments and extensive appendix.

---

> ### Author Response · Authors · 2022-11-17
> **To Reviewer Ube5**
>
> We thank the reviewer for pointing out that our method makes quite a lot of sense, and for offering insightful comments and suggestions. In the following we provide responses to the reviewer’s questions one by one.
>
> Q1. Present the limitations of this type of methodologies right after the contributions in the introduction. So, a new reader can quickly understand where and how this methodology can be applied.
> We follow the reviewer’s suggestion by adding a paragraph just after the contributions in our revision: “Our methodology has limitations (i.e., cases for which performance improvement is less): 1. Dense network, where a variable is coupled with many driving variables such that their causal effects overlap and are harder to distinguish. 2. Intense noise, which makes the casual attention mechanism falsely identify high transfer entropy regions. The downstream classifier then extracts non-causal features, leading to the reduction of its generalization. 3. Strongly coupled system, which is dominated by synchronization phenomena in which the dynamic behaviors of all variables are similar”.
>
> Q2. Regarding the possibility of inferring the informative region by understanding the bifurcations of the individual neuron dynamics.
> We agree with the reviewer that the largest information gains will happen at the position where child neuron escapes away of resting state if one can understand the bifurcations of neuronal dynamics. However, for real systems, the underlying dynamics are usually unknown hence it is difficult to perform bifurcation analysis based on governing equations. Therefore, here we propose the causal attention mechanism to identify the region, where perturbations will lead to the large information gains, from observed neuronal activities. In revision we describe this motivation in the first paragraph of Introduction. We also agree with the reviewer that incorporating the prior knowledge of dynamics (if we could get) to the models (e.g., physic-informed machine learning) will provide more scientific insights and probably yield more accurate causal inference, which is a very interesting direction for future work.
>
> Q3. Regarding the balance between learning rates and the attention variable.
> Indeed, such a balance is important. In our manuscript we take the following into consideration. First, the learning rate for attention model needs to be small (which was set to 10e-5 in all our experiments). A large learning rate will probably make the model ignore the subtle and sparse perturbation effects and assig zero attention coefficient throughout the whole time series. Second, there exists an optimal loss interval for training the attention model. During the training process, the first stage of our method will stop when the loss function Eq.10 is stable in this interval, then the downstream classifier usually obtains the best generalization performance. This optimal interval is different for different dynamics, and we found them empirically. Joint training of Eq.10 and Eq.11 is an alternative design that obtains a better balance between objectives Eq.10 and Eq.11 automatically, which we will try in the next work. In revision we added these considerations in Sec. 3.3.
>
> Other changes:
> (1) In the revision we noted that the added noise is measurement noise on observed data rather than the intrinsic stochasticity of dynamical systems. According to the description in the comment, we speculate the intrinsic stochastic noise that is in the same direction (both positive or both negative) with deterministic dynamics itself will helps to enhance the amplified perturbation effect nearby the unstable saddles. We are interested in examining its influence.
>
> (2) “Why are information from parent...?” – In the revision we changed the sentence to: “the model only exploits the historical information of the child variable itself and that from parent variables is ignored”.
>
> (3) We changed the name of our method to 'Attentive Transfer Entropy' (ATEn) to avoid the abbreviation confusion between ‘Average Treatment Effect’ and our abbreviation.

---

### Official Review · Reviewer_pNYJ · 2022-10-24

**Confidence:** 3
**Correctness:** 3
**Technical Novelty And Significance:** 3
**Empirical Novelty And Significance:** 3
**Recommendation:** 5

**Clarity, Quality, Novelty And Reproducibility:**

Clarity:  The method part (the deriving) is clear. But the insight into the method is unclear. Besides the title and some claims should be refined.

Novelty.  The difference between the proposed method and the original TE method is clear.   The performance improvement is significant with this simple modification. But more details should be explained, such as why this method works well.

Reproducibility: code is attached.

**Strength And Weaknesses:**

Strength:
1) Identifying causal relations of time-series data plays an important role in AI applications, such as science and human-robot interaction.
2) Compared with the original methods using transfer entropy to detect causal-effect relations, this paper learns a reweighted transfer entropy to refine the construction.
3)  This method shows great performance in the experiments.
Weaknesses:
1) "Causal attention" is not accurate. This paper applies a reweighted transfer entropy to promote the construction but does not learn an attention model for causal structure prediction. It is suggested to modify the title and claims in the paper. For example, detect causal effects in time-series data with reweighted/attentive transfer entropy.
2) What is the advantage of reweighting? The insight into using reweight is not clear. Why add a few parameters to reweight can improve the performance dreamily?
3) Is the model identifiable? If yes, it is better to give corresponding proof and assumptions.
4) Could the author provide a detailed ablation study? It seems to be important to identify which component is the key to performance improvement.
5) Could this method be used for i.i.d data? If not, the boundary of the method should be clarified.



**Summary Of The Paper:**

This paper proposes a causal attention model to identify the causal effects in the time series data. This model is trained by maximizing the Attention-extended Transfer Entropy. Beyond the causal attention model, a binary classification module is introduced to mitigate the negative effect of noises. The experiments on both synthetic and real datasets show good performance.


**Summary Of The Review:**

Overall, after balancing the positive and negative points, I think this paper needs to be further polished since many details need to be explained.

---

> ### Author Response · Authors · 2022-11-10
> **To Reviewer pNYJ**
>
> To Reviewer pNYJ
>
> We thank the reviewer for highlighting that the derivation of our methodology is clear. We would like to answer the reviewer’s questions especially about why our method is advantageous.
>
> Q1. What is the advantage of reweighting?
> The main challenge in our task of network inference is that the causal effect is sparse in a class of nonlinear coupling dynamics that are common in neuroscience. The sparse causal effect means the coupling force from $X_{i}$ to $X_{j}$ is observably larger than zero (i.e., has influence on the evolution of time series of $X_{j}$) only momentarily, as illustrated in Figure 3 (these sparse spikes in gray lines, for example, originate from the firing behavior of driver neurons). Thus, even if two variables have a coupling relationship, they seem to be independent of each other at most times. This dilutes the evidence about causal effect (which emerges only in special critical regions) in the time series of $X_{j}$ that can be used to infer the coupling relationship from $X_{i}$ to $X_{j}$.
>   The traditional methods (baselines) that infer the causal effect throughout the whole time series, for which all positions in the time series are treated equally, may essentially ignore these sparse causal effects. More importantly, because of the lacking prior knowledge of the dynamical equation, we don’t know where these causal effects from $X_{i}$ exist in time series of $X_{j}$. To solve this challenge, we propose the causal attention mechanism which uses a machine learning model to autonomously identify the critical regions in time series where transient causal effects may emerge. The model is trained by maximizing the attention-extended transfer entropy, i.e., reweighting the transfer entropy. The trained model can identify the high transfer entropy regions in time series which correspond to substantial causal effect (where information of parent variable reduces the uncertainty of child variable due to significant coupling force). In figure 3, we can see that the orange lines keep in step with the gray lines (model gives large weight/attention coefficient in the regions of the spikes in the gray lines).
>   Finally, we note that Reviewer Ube5 kindly provides further motivation of our method, from a dynamical systems perspective.
>
> Q2. Give corresponding proof and assumptions.
> The core of our method is to reweight the transfer entropy by neural networks to identify these critical regions autonomously. Therefore, we need to make the objective function of estimating transfer entropy differentiable, and then we can use the gradient algorithm for training. We derived the transfer entropy as the difference between two types of mutual information. The theoretical guarantee of mutual information estimation (Eq. 2) by neural networks provides a theoretical guarantee of transfer entropy estimation by neural networks. In Appendix B, we offered a theoretical proof for the consistency and convergence properties of Transfer Entropy Neural Estimation, and examine its bias on a linear dynamic system where the true values of transfer entropy can be determined analytically. Please see also the last paragraph of Sec. 3.1.
>   Our assumption is that the evolution of time series on variables is governed by a general differential dynamical equation. In our revision, we will emphasize this in the first paragraph of Sec. 1.
>
> Q3. Which component is the key to performance improvement.
> There are two steps in our method: causal attention mechanism identifies the critical regions, and the binary classifier makes causal relationship inference focusing on these regions. We consider two baselines: original transfer entropy and a binary classifier with traditional attention mechanism. The traditional attention mechanism is designed for computer vision tasks and cannot catch the causal effect (see light pink lines in Figure 3, these remain close to their maximum value of 1 and are insensitive to changes in coupling force). This leads its classifier extracts feature throughout the whole time series (instead of focusing on causal features), and obtains low loss on training set (spurious statistical correlation) but does not generalize well to test sets. The estimated value of the original transfer entropy is also dominated by the majority of positions where causal effect is negligible. Thus, identification of critical regions in the time series is the key to performance improvement. For additional evidence, in the experiment on real networks, we add intensive noise on time series (to simulate reality) resulting in identifying false high transfer entropy regions that have nothing to do with causal effects. Then, the improvements decrease more than in synthetic networks when critical regions were identified accurately. In our revision we explain more clearly the crucial role of our attention mechanism, for example, in Sec. 4.2.

---

> > ### Author Response · Authors · 2022-11-10
> > **To Reviewer pNYJ**
> >
> > Q4. Concept of ‘attention’.
> > Our method is consistent with our understanding of attention in machine learning, which is analogous to that “attention mechanisms identify key areas in the data by learning a set of weight distributions” presented in, e.g. Ref. [1][2] immediately below. Specifically, as described in Section 3.3, we employ a neural network (multilayer perceptron trained by loss function Eq. 10) $g_{\alpha }$ to assign the weight/attention coefficients throughout the time series together with the transfer entropy estimator $f_{\theta}$ and $f_{\phi}$ to find the high transfer entropy regions. We hope it is acceptable to continue using our preferred title.
> >
> > [1] Hassanin, M., Anwar, S., Radwan, I., Khan, F. S., & Mian, A. (2022). Visual Attention Methods in Deep Learning: An In-Depth Survey. arXiv preprint arXiv:2204.07756.
> > [2] Woo, S., Park, J., Lee, J. Y., & Kweon, I. S. (2018). Cbam: Convolutional block attention module. In Proceedings of the European conference on computer vision (ECCV) (pp. 3-19).
> >
> >
> > Q5. The boundary of the method should be clarified.
> > The applications of our method have limitations: a) Dense network. b) Intensive noise. c) Strong coupling. In our revision, we detail these limitations in last paragraph of Sec. 1.
> >
> > We hope that our response clarifies the main challenge in our task, and how the challenge motivates our innovation.

---

> > > ### Comment · Reviewer_pNYJ · 2022-11-19
> > > **Response to Authors**
> > >
> > > Dear Authors,
> > >
> > > I greatly appreciate the author's detailed point-to-point responses. However, my concerns are not addressed well.
> > >
> > > 1) I agree with the Reviewer F8Hs, it is still unclear how this paper contributes to the existing literature in the revised paper.
> > >
> > > 2) Attention.  The definition of attention is totally right. My concern is about "causal attention".  For example, when we name a "new attention", we should clarify the difference between this attention and other attention mechanisms, like that  "Cross attention" introduces the interactions of two domains than self-attention. What is the advantage of "causal attention" over typical attention?  As noticed in the feedback, the attention in this paper is the same as conventional ones, such as using attention (reweight）on transfer entropy.
> > >
> > > 3) Identifiability.  When we learn causal relations, their identifiability is necessary.  I don't find any response to this question.
> > >
> > > I will calibrate my evaluation after further feedback. Just for now,  I tend to reduce my score since some key concerns （expected to be resolved） are not addressed.
> > >
> > > Many thanks,
> > >
> > > Reviewer pNYJ

---

> > > > ### Author Response · Authors · 2022-11-20
> > > > **Further Response to Reviewer pNYJ**
> > > >
> > > > Dear Reviewer pNYJ,
> > > >
> > > > We are delighted receiving your further feedback and really appreciate that you clarified your remaining concerns. We followed your insightful suggestions, which are very helpful for us to further improve our manuscript. Sadly, the track for resubmitting a revised manuscript is closed now. In the following we describe our revision and will make the revision accordingly in the next-round submission.
> > > >
> > > > 1. The contribution of our work: Because the causal effect in neuronal dynamics data is usually transient and weak, i.e. having effect only during short time intervals instead through the whole time series, in this work we draw on the attention mechanism to identify these critical regimes for inferring such subtle causal effects. We revised the description of our contribution in Abstract as ‘Here we draw on the attention mechanism to guide the classifier to make inference focusing on the critical regions of time series data where causality may manifest’, and in Introduction-contribution summary as ‘We draw on the attention mechanism for causal relation inference by identifying…’
> > > >
> > > >
> > > > 2. Attention: We feel sorry that we had not totally understood your concern previously. Now we totally get it and have revised our manuscript accordingly:
> > > >
> > > > (2.1) We rephrased our contribution in Abstract and Introduction as described above, changing ‘propose a causal attention mechanism’ to ‘draw on the attention mechanism …’.
> > > >
> > > > (2.2) We followed your suggestion and changed our title to ‘Attentive Transfer Entropy to Exploit Transient Emergence of Causal Effect in Time-Series Data’. Indeed, ‘Attentive Transfer Entropy’ is a wonderful name for our model and emphasizing ‘time-series data’ is helpful for the readers to easily understand the problem we are dealing with. Thank you! Yet, we prefer to keep ‘Transient Emergence of Causal Effect’ in the title as it is the core motivation and a peculiarity of our work.
> > > >
> > > > (2.3) We added a subsection ‘Related Work - 5.3 Attention Mechanism’ : ‘The attention mechanisms identify key areas in the data by learning a set of weight distributions. Spatial-based attention [Woo et al. (2018); Hsieh et al. (2019); Shen et al. (2020)] involves generating attention scores from spatial regions of feature maps, while channel-based attention [Hu et al. (2018); Zhang et al. (2020)] optimises the representation of each channel. Self-attention [Vaswani et al. (2017); Wang et al. (2018)] encodes interactions among all input entities and cross-attention [Li et al. (2021); Chen et al. (2021); Jaegle et al. (2022)] introduces the interaction of two domains further. However, it is recognised that attention mechanisms need to be tailored to the specific problem at hand [Hassanin et al. (2022)]. In our work, we tailor attention mechanism for causal relation inference by accommodating the selection of temporal regions that correspond to the transient causal effect in time series.’
> > > >
> > > > 3. Identifiability. a) Causal graphs with the same $d$ separation structure and the same conditional independence are Markov equivalence class, and the causal direction cannot be distinguished according to the conditional independence. This is a classic problem of identifiability. For example, $X \to Y \to Z$, $X \gets Y \gets Z$ and $ X \gets Y \to Z $. They are $X \perp Z \parallel Y$. However, for the causal inference based on time-series data with delay and autoregression (to which our tasks belong), e.g., $X_{t- 1 }^{i} \to X_{t}^{i}, Y_{t- 1 }^{i} \to Y_{t}^{i}, X_{t- \tau }^{i} \to Y_{t}^{i}$, the direction between $X$ and $Y$ can be determined by exploring the asymmetry of conditional independence test that dependency $X_{t}^{i}$ and $Y_{t}^{i}$ conditioned on $ Y_{t- \tau }^{i} $ > that conditioned on $ X_{t- \tau }^ {i}$. Once the direction between $X$ and $Y$ is determined, the causal structure in the class mentioned above can be distinguished. This is also the reason of that there have been increasing interest in time-series-based causal graph inference in recent years. However, the problem we aim to solve in the paper is more challenging because of the weak and sparse causal effect in neural dynamics, which causes that two time-series may appear statistically independent even if they have the causal relationship.  b) Practically, recent methods give out the score between $\[0,1\]$ (0\1 represents non-causal\causal relationship) for each pair of variables rather than candidate causal graphs. Compared with true labels on pairs, their performance can be evaluated by metrics for binary classification.
> > > >
> > > > We enjoyed the discussion with you, and thank you again for your timely feedback.
> > > >
> > > > Best,
> > > >
> > > > Authors

---

### Official Review · Reviewer_F8Hs · 2022-10-25

**Confidence:** 3
**Clarity, Quality, Novelty And Reproducibility:** Please see my response above.
**Correctness:** 3
**Technical Novelty And Significance:** 2
**Empirical Novelty And Significance:** 2
**Recommendation:** 5

**Strength And Weaknesses:**

This paper may have some interesting ideas. The definition of attention-extended transfer entropy (ATE) seems intriguing. However, due to the lack of clarity, it is unclear how this paper contributes to the existing literature and how novel the proposed method is. I will elaborate on the following.

First, the authors claim that "our task is to infer causal relationships between observed variables based on time series data and reconstruct the causal network connecting large numbers of these variables." However, this inference task is never formally defined. What is the performance measure of the learning task? How does one measure the quality of the reconstructed causal networks? Should we measure the L1 / L2 distance of the learned parameters with the actual parameters of the underlying model, or should we measure of the divergence between simulated and observed samples? Unfortunately, I tried to read through the paper but could not find answers.

Second, the paper describes the proposed algorithm. However, it does not perform any analysis of the algorithm's theoretical guarantee, concentration properties, and sample complexity. It is unclear how the proposed algorithm improves the existing baseline. To support the proposed method, the authors have to resort to empirical evaluation.

As for the experiments, I appreciate the authors' efforts in including various synthetic and dynamical models. Unfortunately, the clarity issue remains. For instance, the authors state "compared with the baselines, our method usually substantially improves
reconstruction performance on both synthetic and real causal networks, as shown in Figure 4." Again, it is unclear how the reconstruction performance is measured here. Without such information, it is hard to evaluate and compare the proposed method with the existing baseline.

**Summary Of The Paper:**

This paper studies a cyclic causal model where values of every variable $X_i$ at time step t are decided by a differential equation $\frac{d x_i}{d x}  = g(x_i) + \sum_{j} B_{ij}f(x_i, x_j)$. The learner's goal is to recover the structural functions $f$ and $g$ and the coefficients $B_{ij}$ from the observational data. The authors propose a new loss function for this learning task, called attention-extended transfer entropy, which encourages certain conditional dependence in the model. Finally, simulations were performed on both synthetic and dynamical models to validate the proposed approach.

**Summary Of The Review:**

This paper studies a cyclic causal model where values of every variable $X_i$ at time step t are decided by a differential equation $\frac{d x_i}{d x}  = g(x_i) + \sum_{j} B_{ij}f(x_i, x_j)$. The learner's goal is to recover the structural functions $f$ and $g$ and the coefficients $B_{ij}$ from the observational data. The authors propose a new loss function for this learning task, called attention-extended transfer entropy, which encourages certain conditional dependence in the model. However, this paper provides little theoretical guarantee for the proposed method. Due to the lack of clarity, it is unclear how this paper contributes to the existing literature and how novel the proposed method is.

---

> ### Author Response · Authors · 2022-11-08
> **To Reviewer F8Hs**
>
> We thank the reviewer for pointing out that the definition of attention-extended transfer entropy is intriguing. We read carefully the reviewer’s comments and realized that we might not communicate clearly the purpose of our work. Hence, we would like to offer a point-by-point response, improving the clarity, and wish that the reviewer reconsiders our work.
>
> Q1. Our goal.
> Our goal is not to infer the functions $f$ and $g$ and their coefficients, but to recover the topological structure of the causal network, which can be represented as an unweighted asymmetric adjacent matrix $B_{ij}$. When $X_{i}$ is a coupling driver (parent variable) of $X_{j}$, the element in matrix $B_{ij}$ (i.e., coupling relationship) is 1, otherwise it is zero. The asymmetry means this causal network is a directed network. Hence, in the first paragraph of Sec. I, we described our inference task: Given unknown functions $f$ and $g$, we aim to infer the causal network (matrix $B_{ij}$) from observed nodal activities $X_{i}$ , i=1,2,…,N. In our revision we will further improve the clarity of the description of our inference task, trying to avoid any potential confusion.
>
> Q2. Performance measure.
> For each ordered pair of variables (each element in $B_{ij}$) our task it is a binary classification problem (0 or 1). If $X_{i}$ is a coupling driver of $X_{j}$, a good algorithm should take the time series as input and infer that $B_{ij}$ is 1 rather than 0. The more coupling relationships between variables are correctly inferred, the more accurately the causal network can be reconstructed. Hence, as described in Sec. 4.1, we used the AUROC and AUPRC to evaluate the performance of our algorithm, which are two common and effective metrics for the binary classification problem.
>
> Q3. How novel the proposed method is.
> The main difficulty of the inference task we aimed to solve is the sparseness of causal effect in neural dynamics, i.e., the coupling force from $X_{i}$ to $X_{j}$ is observably larger than zero (has influence on the evolution of time series of $X_{j}$) only momentarily, as illustrated in Figure 3. Thus, even if two variables have a coupling relationship, they seem to be independent of each other at most times. This dilutes the evidence about causal effect (which emerges only in special critical regions) in the time series of $X_{j}$ that can be useful to infer the coupling relationship from $X_{i}$ to $X_{j}$.
> Our innovation is to propose the causal attention mechanism to catch this sparse evidence autonomously by a machine learning model trained using the objective of the attention-extended transfer entropy. This mechanism guides a binary classifier to make inference focused on these critical regions, as described in the third and fourth paragraphs of Sec. 1. By contrast, traditional causal detection methods that infer the causal effect throughout the whole time series may ignore these sparse causal effects. Our results showed that the identification of critical regions in time series indeed significantly improves the performance of our algorithm compared with baselines.
> We hope that we have clarified the innovation and contribution of our work, especially for causal inference problems in coupling dynamics systems where coupling force is weak and sparse.
>
> Q4. Theoretical guarantee for the proposed method.
> The core of our method is to reweight the transfer entropy by a neural network to identify the critical regions in time series where transient causal effects may exist. Therefore, we need to make the objective function that transfer entropy differentiable. We derived the transfer entropy as the difference between two types of mutual information. The theoretical guarantee of mutual information estimation by neural networks provides a theoretical guarantee of transfer entropy estimation by neural networks.
> In Appendix B, we offered a theoretical proof for the consistency and convergence properties of Transfer Entropy Neural Estimation, and examine its bias on a linear dynamic system where the true values of transfer entropy can be determined analytically. Please see also the last paragraph of Sec. 3.1.
>
> In sum, we wish that the improvements to our presentation which the reviewer has kindly facilitated make clearer our work’s motivation, methods, theoretical basis, novelty, and contribution.

---

> > ### Comment · Reviewer_F8Hs · 2022-11-25
> > **Thanks for the response**
> >
> > I thank the authors for the provided feedback. I have read it together with the other reviews. Some of my concerns have been addressed. But it also raises more questions. For instance, the authors claim, "our goal is not to infer the functions f and g and their coefficients, but to recover the topological structure of the causal network." In this case, it is somewhat surprising that several causal discovery algorithms are not included as baselines. The authors state that "many methods of causal discovery assume that the causal network is a directed acyclic graph." This might be true for earlier work, but more recently, causal discovery algorithms have been extended to time series data, e.g. (Entner & Hoyer, 2010; Malinsky & Sprites, 2018). The contributions of this paper are somewhat unclear without comparing them to existing methods.
> >
> > Also, the proposed algorithm outputs a specific instance of a causal model with a fixed causal structure. This is curious since it is well-known that the structure of the underlying causal graph is not identifiable from the observational data. In this case, existing causal discovery methods are generally conservative and return a class of causal graphs that potentially generate observational data. The structure of these candidate causal graphs could be significantly different, which leads to distinct conclusions in their generality, e.g., inferring a causal effect. In this sense, I am not confident in the soundness of the proposed method without a rigorous theoretical grounding.

---

> > > ### Author Response · Authors · 2022-11-29
> > > **Further Response to Reviewer F8Hs**
> > >
> > > Dear Reviewer,
> > >
> > > Thanks for your feedback! We are glad that most previous concerns have been addressed. For the two additional questions, we would like to offer a point-to-point response as follows.
> > >
> > > Q1. Regarding baselines.
> > > As shown in Table 3 and Sec. 5.1 we have considered 6 widely-used time series based causal graph reconstruction methods as baselines. Regarding the method you mentioned in the review report, i.e., Fast Causal Inference (FCI, Entner & Hoyer, 2010; Malinsky & Sprites, 2018), we actually compared with its offspring PCMCI$^{+}$. Previous studies, e.g., J. Runge, UAI 2020, showed that PCMCI$^{+}$ outperforms FCI for time series with strong autocorrelation (just like in our tasks, where the self-drive force dominates the evolution of variables). Therefore, our approach should outperform FCI too. Nevertheless, we follow your suggestion and will add FCI (Entner & Hoyer, 2010) into the comparisons in the next-round submission.
> > >
> > > Q2. Regarding identifiability.
> > > Thank you for the insightful comment. As we know, causal graphs with the same $d$ separation structure and the same conditional independence are Markov equivalence class, and the causal direction cannot be distinguished according to the conditional independence. For example, $X \to Y \to Z$, $X \gets Y \gets Z$ and $ X \gets Y \to Z $. They are $X \perp Z \parallel Y$. However, for the causal inference based on time-series data with delay and autoregression (to which our tasks belong), e.g., $X_{t- 1 }^{i} \to X_{t}^{i}, Y_{t- 1 }^{i} \to Y_{t}^{i}, X_{t- \tau }^{i} \to Y_{t}^{i}$, the direction between $X$ and $Y$ can be determined by exploring the asymmetry of conditional independence test that dependency $X_{t}^{i}$ and $Y_{t}^{i}$ conditioned on $ Y_{t- \tau }^{i} $ > that conditioned on $ X_{t- \tau }^ {i}$. Once the direction between $X$ and $Y$ is determined, the causal structure in the class mentioned above can be distinguished. This is also the reason of that there have been increasing interest in time-series-based causal graph inference in recent years. Actually, the problem we aim to solve in the paper is more challenging because of the weak and sparse causal effect in neural dynamics. Our results showed that, adding attention mechanism into time-series-based causal inference indeed significantly improves accuracy. We believe that our contribution will open new possibilities for solving this long-standing problem.
> > >
> > > We hope that our response and future corresponding revision have addressed your additional concerns. Thank you again!
> > >
> > > Best,
> > > The Authors

---

### Decision · Program_Chairs · 2023-01-20

**Decision:**

Reject

**Justification For Why Not Higher Score:**

Two reviewers still have concerns about clarity, baselines, ablation studies, and lack of analysis and theoretical guarantees (identifiability).

**Justification For Why Not Lower Score:**

n/a

**Metareview: Summary, Strengths And Weaknesses:**

Reviewers have rated the paper 5 (F8Hs), 5 (pNYJ), and 8 (Ube5). A decent amount of discussion has taken place. As per their last messages it seems that F8Hs and pNYJ still have concerns regarding clarity, baselines, ablation studies, and lack of analysis and theoretical guarantees. Importantly, there is no guarantee on identifiability, which seems to me to be necessary to justify the use of the word "causal".

**Summary Of Ac-Reviewer Meeting:**

I tried to set up a meeting, but no reviewer responded to my request